# Copper sulfide nanoparticles as a photothermal switch for TRPV1 signaling to attenuate atherosclerosis

Wen Gao[1], Yuhui Sun[1], Michelle Cai[2], Yujie Zhao[1], Wenhua Cao[1], Zhenhua Liu[1], Guanwei Cui[1] & Bo Tang[1]

Atherosclerosis is characterized by the accumulation of lipids within the arterial wall. Although activation of TRPV1 cation channels by capsaicin may reduce lipid storage and the formation of atherosclerotic lesions, a clinical use for capsaicin has been limited by its chronic toxicity. Here we show that coupling of copper sulfide (CuS) nanoparticles to antibodies targeting TRPV1 act as a photothermal switch for TRPV1 signaling in vascular smooth muscle cells (VSMCs) using near-infrared light. Upon irradiation, local increases of temperature open thermo-sensitive TRPV1 channels and cause $Ca^{2+}$ influx. The increase in intracellular $Ca^{2+}$ activates autophagy and impedes foam cell formation in VSMCs treated with oxidized low-density lipoprotein. In vivo, CuS-TRPV1 allows photoacoustic imaging of the cardiac vasculature and reduces lipid storage and plaque formation in $ApoE^{-}/^{-}$ mice fed a high-fat diet, with no obvious long-term toxicity. Together, this suggests CuS-TRPV1 may represent a therapeutic tool to locally and temporally attenuate atherosclerosis.

[1] College of Chemistry, Chemical Engineering and Materials Science, Collaborative Innovation Center of Functionalized Probes for Chemical Imaging in Universities of Shandong, Key Laboratory of Molecular and Nano Probes, Ministry of Education, Institute of Biomedical Sciences, Shandong Normal University, Jinan 250014, China. [2] Faculty of Science, Western University, London, ON N6A 3K7, Canada. Wen Gao and Yuhui Sun contributed equally to this work. Correspondence and requests for materials should be addressed to B.T. (email: tangb@sdnu.edu.cn)

Atherosclerosis is the leading cause of cardiovascular and cerebrovascular events[1]. The key event in early atherosclerosis is cholesterol and triglycerides accumulation in vascular smooth muscle cells (VSMCs) and monocytes/macrophages, leading to the formation of foam cells[2, 3]. Recently, transient receptor potential vanilloid subfamily 1 (TRPV1), a thermosensitive cation channel that is activated by capsaicin, has been found to protects against foam cell formation through inducing autophagy in oxidized low-density lipoprotein (oxLDL)-treated VSMCs[4]. Autophagy is a reparative, life-sustaining process by which cytoplasmic oxLDL is sequestered in double membrane vesicles and delivered to lysosomes after fusion with lysosomal compartments. Upon delivery, lysosomal acid lipase acts to hydrolyze oxLDL to generate free cholesterol mainly for ATP-binding cassette transporter A1 (ABCA1)-dependent efflux[5–7]. Therefore, activation of TRPV1 signaling pathway is an attractive mean to reduce lipid accumulation and VSMC foam cell formation. Although this may be a promising therapeutic target for atherosclerosis, direct use of capsaicin as a TRPV1 agonist in clinical applications is limited by its toxic side effects, such as skin irritation, persistent desensitization and cocarcinogenic effect[8, 9]. Moreover, capsaicin is diffusible and unable to regulate TRPV1 signal transduction in a controlled manner, which can act outside of the confines of the atherosclerotic plaques. Being able to manipulate TRPV1 signaling at precise times and spaces in living systems remains a challenge.

Nanoparticles (NPs) have emerged as a powerful tool for controlling cell signaling pathways with high spatial and temporal resolution[10–14]. Owing to their unique physical and chemical properties, optical, electrical and magnetic methods have been devised to regulate cell signaling[15–18]. Among these, optical stimuli, especially using near-infrared (NIR) light, is uniquely advantageous because it can penetrate deeply with negligible attenuation into biological tissues and minimal photodamage to cells[19]. Aiming to activate TRPV1 signaling using NIR light, we turned our attention to the characteristic NIR absorption of copper sulfide (CuS) NPs. Unlike the optical absorption in gold nanostructures and carbon nanotubes based on the surface plasmon resonance (SPR)[20, 21], NIR absorption of CuS NPs derives from the d-d transition of $Cu^{2+}$ ions, which is not affected by the solvent or the surrounding environment when formulated or delivered in vivo[22, 23]. Irradiation of CuS NPs by NIR results in local heating and generates strong photoacoustic (PA) signal[24, 25]. PA imaging is a high-resolution optical imaging modality used to visualize blood vessels in deeper regions[26]. Therefore, CuS NPs are promising for gating of the thermosensitive TRPV1 ion channel as well as for PA image-guided therapy of atherosclerosis.

Here, we develop a CuS NPs-based switch for photothermal activation of TRPV1 signaling to impede the progression of atherosclerosis (Fig. 1). This switch consists of CuS NP conjugated with a TRPV1 monoclonal antibody (CuS-TRPV1), which enables specific binding to TRPV1 on the plasma membrane of VSMC. Following NIR laser irradiation, the increase in local temperature opens TRPV1 channels allowing an influx of calcium ions ($Ca^{2+}$). The increased cytosolic $Ca^{2+}$ leads to subsequent autophagy activation, which upregulates ABCA1-mediated cholesterol efflux and reduces lipid accumulation and foam cell formation in oxLDL-treated VSMCs. Importantly, CuS-TRPV1 is able to provide obvious structural PA imaging of cardiac vasculature, making it feasible for precise temporal and spatial control of TRPV1-signaling in vivo. After 12 weeks of PA image-guided therapy, lipid storage and atherosclerotic lesions are significantly reduced in aortic arch of apolipoprotein E knockout (ApoE$^{-/-}$) mice on a high-fat diet without noticeable in vivo long-term toxicity. These results greatly motivate the application of CuS-TRPV1 as a therapeutic tool to attenuate

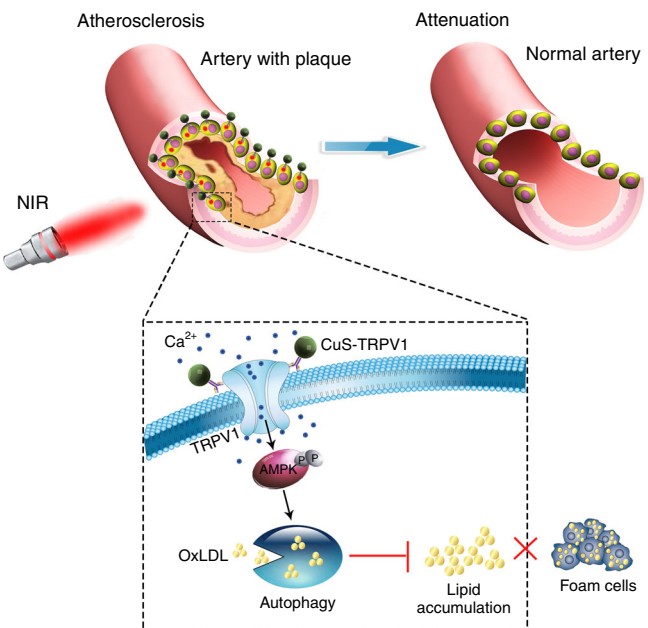

**Fig. 1** Illustration of CuS-TRPV1 switch for photothermal activation of TRPV1 signaling to attenuate atherosclerosis

atherosclerosis through photothermal activation of the TRPV1 signaling pathway.

## Results

**Preparation and characterization of CuS-TRPV1.** To prepare the CuS-TRPV1 switch, citrate-capped CuS NPs (CuS-Cit, 11 ± 2.6 nm, Fig. 2a) were chosen owing to their strong NIR optical absorption and high molar extinction coefficient ($2.6 \times 10^{7}$ cm$^{-1}$ M$^{-1}$ at 1064 nm, Fig. 2b), which is essential for effective utilization of photothermal and photoacoustic effect. TRPV1 antibody is then conjugated to CuS-Cit by means of amide condensation reaction (Supplementary Fig. 1), resulting in a slight increase in particle size (13 ± 1.2 nm) and a red-shift of the absorption peak (1057 nm). The successful fabrication of CuS-TRPV1 was confirmed by sodium dodecyl sulfate polyacrylamide gel electrophoresis (SDS-PAGE, Supplementary Fig. 2) and fourier transform infrared spectroscopy (FT-IR, Supplementary Fig. 3). To evaluate the photothermal conversion, CuS-TRPV1 dissolved in PBS at different concentrations was irradiated with a continuous wave (CW) laser at 980 nm for 350 s and temperature was recorded online. Significant heating of the solution was observed, especially at a high-concentration and a high-power density (Supplementary Fig. 4). CuS-TRPV1 (0.4 mg mL$^{-1}$, 980 nm laser, 5 W cm$^{-2}$) demonstrated a rapid temperature increase from 19.9 °C to 65.8 °C in 180 s (Fig. 2c). By repeating laser on-off cycles, we were able to raise the temperature at each step without compromising the photothermal efficiency (Supplementary Fig. 5), suggesting a potential for remote activation of TRPV1 channels by means of CuS-TRPV1 heating. PA signals of CuS-TRPV1 at 980 nm were measured as shown in Fig. 2d. As expected, the signal intensity was highly dependent on the concentration of CuS-TRPV1.

**In vitro photothermal switching on of TRPV1 signaling.** To effectively trigger the opening of TRPV1 channels on VSMCs, a high local density of NPs was required to cause significant regional heating along the membrane surface. We successfully achieved this by targeting CuS-TRPV1 to VSMCs via a specific monoclonal antibody interaction. Transmission electron

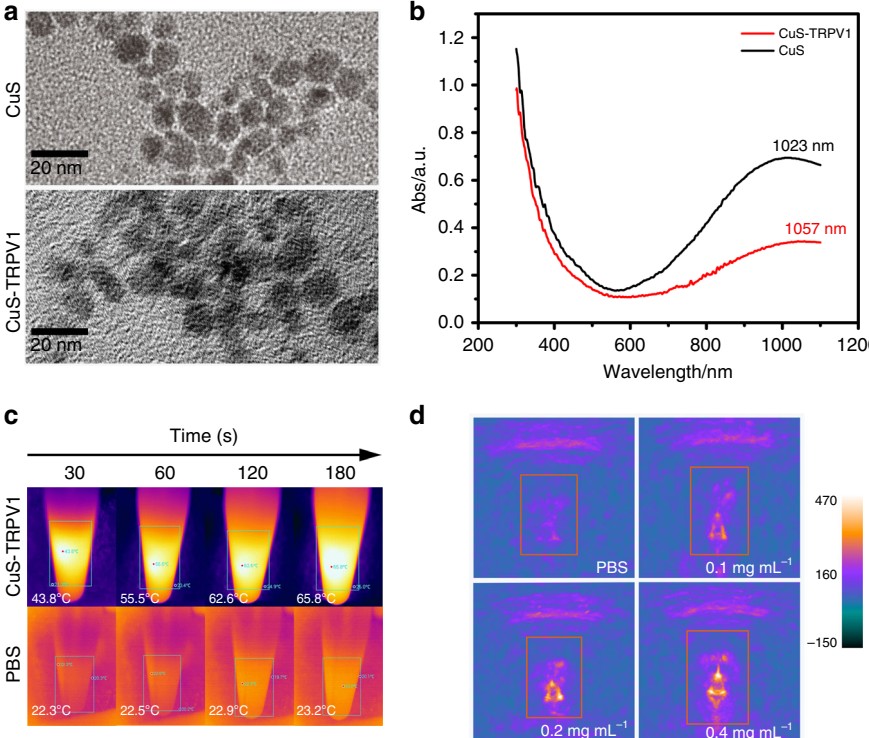

**Fig. 2** Characterization of CuS-TRPV1 switch. **a** Transmission electron micrographs and **b** UV-Vis-NIR spectra of CuS and CuS-TRPV1. Scale bar = 20 nm. **c** Real-time infrared thermal imaging of CuS-TRPV1 and PBS with NIR irradiation (0.4 mg mL$^{-1}$, 980 nm, 5 W cm$^{-2}$ for 180 s). **d** PA imaging of CuS-TRPV1 dispersed in PBS buffers with different concentrations (0, 0.1, 0.2, 0.4 mg mL$^{-1}$)

microscopy (TEM) and imaging flow cytometry (IFC) confirmed that a high density of CuS-TRPV1 was confined to the plasma membrane after 2 h incubation. In contrast, non-coated CuS NPs did not show such specific localization on VSMC membranes, as they were internalized into the interior of cells and accumulated in endosomes or lysosomes (Fig. 3a, b). To investigate the binding affinity of CuS-TRPV1, a competitive binding study was developed based on confocal z-height imaging. TRPV1 channels on VSMCs were first blocked by excess amount of TRPV1 antibody (0.5 mg mL$^{-1}$) and CuS-TRPV1 was subsequently incubated with cells. After confirming the cell focusing ranges from bottom to top, focal plane images were captured at 0.5, 2, 3.5, and 5 μm from the bottom (Supplementary Fig. 6). Without TRPV1 blocking, CuS-TRPV1 bound with TRPV1 on VSMCs within 2 h, showing strong green fluorescence at focal plane near the cell surface (5 μm from the bottom). In contrast, with TRPV1 blocking, only minimal fluorescent signal was observed when we focused at the surface of these cells. These results suggested that CuS-TRPV1 has a specific and strong binding ability with TRPV1 on VSMCs, therefore, resulting in efficient local heating. After 30 s exposure to the laser (980 nm, 5 W cm$^{-2}$), significant temperature increase from 37 to 42.7 °C was observed in the VSMCs incubated with 0.4 mg mL$^{-1}$ of CuS-TRPV1 (Supplementary Fig. 7), which was high enough to activate TRPV1. In the following experiments that tested the remote activation of the TRPV1 channels, NIR irradiation was applied for 30 s followed by a 30 s cooling period to maintain the plasma membrane temperature at less than 43 °C. With up to 30 laser on/off cycles, CuS-TRPV1 did not cause apparent cytotoxicity (Supplementary Fig. 8) and still retained its binding affinity (Supplementary Fig. 6).

To study whether CuS-TRPV1-generated local heating is sufficient in triggering the opening of TRPV1 channels, we measured the intracellular Ca$^{2+}$ concentration using Fluo-3, a Ca$^{2+}$-sensitive fluorescent probe[27]. Within 1 cycle of applying NIR irradiation, a strong increase in green fluorescence signal was observed. This increase was found to be caused by Ca$^{2+}$ influx through thermally activated TRPV1 channels, because cells with CuS-TRPV1 switch but without laser, and cells without both switch and laser, did not show any Ca$^{2+}$ influx. In comparison, Ca$^{2+}$ influx into VSMCs was also evoked by capsaicin (Fig. 3c). These data show that NIR irradiation-induced CuS-TRPV1 heating is sufficient in triggering the opening of TRPV1 channels within seconds. To assess whether the increased cytosolic Ca$^{2+}$ can promote extrinsic signaling process, we monitored activation status of AMP-activated protein kinase (AMPK), one of the main intermediate species of the TRPV1 signaling[28]. NIR excitation of VSMCs pre-incubated with CuS-TRPV1 (0.4 mg mL$^{-1}$) for 10 cycles significantly increased AMPK phosphorylation, but no change was observed in the non-coated CuS group. The photothermal activation resembled the activation achieved by direct treatment of cells with capsaicin (Fig. 3d). The involvement of TRPV1 in AMPK activation was further evaluated by using the TRPV1 antagonist, 50-iodo-resiniferatoxin (iRTX)[29]. As shown in Supplementary Fig. 9, iRTX significantly inhibited TRPV1-activated AMPK phosphorylation in CuS-TRPV1-heated VSMCs, demonstrating that the increase in p-AMPK rooted from TRPV1.

After phosphorylation, AMPK can initiate the autophagic process through recruiting downstream autophagy-related proteins to the autophagosome formation site and governing autophagosome formation[30, 31]. We then sought to use CuS-TRPV1 to photothermally induce autophagy, thus controlling the lipid metabolism in oxLDL-treated VSMCs. In the presence of CuS-TRPV1, oxLDL-treated (24 h) VSMCs were irradiated with NIR laser. After 30 cycles, numerous double-membrane structures were revealed by TEM, which is characteristic of autophagosomes (Fig. 4a). The impaired autophagy by oxLDL was markedly rescued by CuS-TRPV1, which was further confirmed by two of the autophagy markers: downregulated

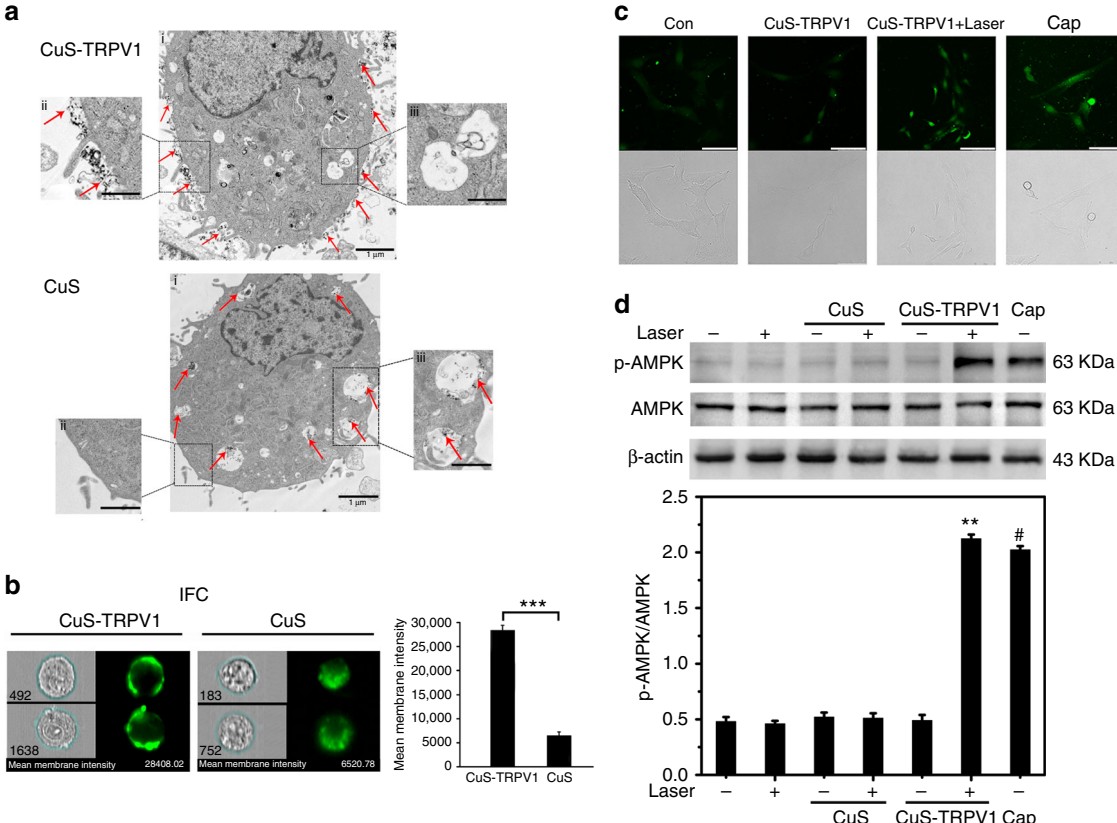

**Fig. 3** Targeting of CuS-TRPV1 to VSMCs membrane and opening of TRPV1 channels. **a** Representative TEM images of VSMCs incubated with 0.4 mg mL$^{-1}$ of CuS-TRPV1 or CuS for 2 h. Red arrows indicated localization of NPs in (i) whole cell section under low magnification, (ii) partially magnified cell membrane and (iii) magnified endosome or lysosome. Images are representative of three independent wells per group. Scale bar = 1 μm. **b** Representative IFC images of VSMCs incubated with the fluorescein-conjugated CuS-TRPV1 or CuS (0.4 mg mL$^{-1}$) for 2 h. Green fluorescence in the blue area indicated NPs targeting to the cell membrane. Mean membrane intensity of CuS (blue area: 6,520.78) was 23% of CuS-TRPV1 (blue area: 28,408.02). Data are shown as mean ± S.D. of three independent experiments, and analyzed by Student's $t$-test. ***$P < 0.001$. **c** Fluorescence imaging of Ca$^{2+}$ influx in VSMCs induced by CuS-TRPV1 heating. VSMCs were incubated with 0.4 mg mL$^{-1}$ of CuS-TRPV1 and irradiated by the 980 nm laser (5 W/cm$^2$) for 1 cycle. Capsaicin (1 μM) was used as a positive control. Images represent three independent wells per group. Scale bar = 50 μm. **d** Western blot analysis of AMPK phosphorylation in VSMCs induced by CuS-TRPV1 heating-evoked Ca$^{2+}$ influx. VSMCs were incubated with 0.4 mg mL$^{-1}$ CuS-TRPV1 and irradiated by the 980 nm laser (5 W cm$^{-1}$) for 10 cycles. Capsaicin (Cap, 1 μM) was used as a positive control and CuS (0.4 mg mL$^{-1}$) was used as a negative control. The relative levels of AMPK phosphorylation were normalized to total AMPK, with β-actin serving as a loading control. Data are shown as mean ± S.D. of three independent experiments, and analyzed by Student's $t$-test. **$P < 0.01$ for CuS-TRPV1 +Laser vs. untreated group. #$P < 0.05$ for Cap vs. untreated group

expression of LC3I and upregulated expression of LC3II[31, 32]. As shown in Fig. 4b, LC3II/LC3I ratio increased in a time-dependent manner, with an obvious effect at 30 cycles post NIR irradiation. Autophagy, through an increase in LC3II/LC3I ratio, also occurred in rapamycin (an inducer of autophagy) and capsaicin treated group, but did not occur when each component of the photothermal switch, such as NIR laser or CuS NPs, was applied alone. The promotion of CuS-TRPV1 on LC3II/LC3I ratio was reversed by iRTX (Fig. 4c and Supplementary Fig. 10). Moreover, activation of autophagy inhibited oxLDL-induced VSMC foam cell formation manifested by decreased lipid droplet accumulation (Supplementary Fig. 11a) and total cholesterol level (Supplementary Fig. 11b). This decrease can be attributed to an upregulation of ABCA1 expression, leading to enhanced cholesterol efflux (Supplementary Fig. 12). These data demonstrated that opening of TRPV1 channels by means of CuS-TRPV1 heating was sufficient to inhibit foam cell formation through rescuing the impaired autophagy and increasing cholesterol efflux in oxLDL-treated VSMCs.

It is worth noting that in contrast to capsaicin which typically induce systemic stimulation of TRPV1 signaling in the whole body[28, 33, 34], the CuS-TRPV1 switch system has the merit

of spatial controllability using the application of NIR irradiation to a selected area. To demonstrate local activation of TRPV1 signaling, a 980 nm NIR laser (5 W cm$^{-2}$) was focused to a micrometer-size area (spot size 300 μm in diameter) of a cell culture slide containing VSMCs by using the NIR irradiation shown in Supplementary Fig. 13a. On application of the NIR irradiation for 30 cycles, activation of autophagy, seen in the form of an immunostained green fluorescence signal (LC3B), was observed, exclusively in the circular area with about 400 μm diameter under the laser (Supplementary Fig. 13b). This testified to the spatial controllability of the CuS-TRPV1 switch.

**In vivo photothermal switching on of TRPV1 signaling.** Having established that this photothermal switching strategy for TRPV1 signaling is effective in living cells, we finally sought to demonstrate the potential application of CuS-TRPV1 at the animal level. The CuS-TRPV1 switch was stable for up to 7 days in serum containing media at 37 °C, exhibiting no significant aggregation and retaining its optical adsorption in the NIR (Supplementary Fig. 14). After incubation with VSMCs for 24 h,

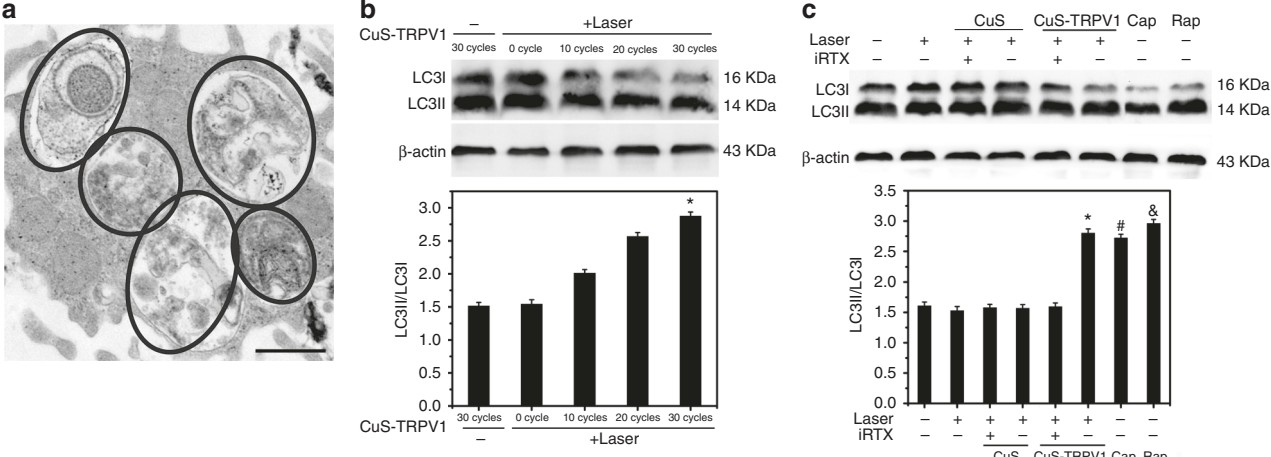

**Fig. 4** Photothermal activation of TRPV1 by CuS-TRPV1 rescued the autophagy impaired by oxLDL in VSMCs. 80 µg mL$^{-1}$ oxLDL-pretreated VSMCs were incubated with CuS-TRPV1 (0.4 mg mL$^{-1}$) and irradiated by the 980 nm laser (5 W cm$^{-2}$). **a** Representative TEM images of autophagosomes in VSMCs after 30 irradiation cycles. Black circles were added to outline the double-membrane structures of autophagosomes. Images are representative of three independent wells. Scale bar = 1 µm. **b** Western blot analysis of LC3II/LC3I ratio at the indicated irradiation cycle. β-actin was used as a loading control. Data are shown as mean ± S.D. of three independent experiments, and analyzed by Student's $t$-test. *$P$ <0.05 vs. 0 cycle. **c** Western blot analysis of LC3II/LC3I ratio in VSMCs after indicated treatments. Capsaicin (Cap, 1 µM) and rapamycin (Rap, 10 nM) were used as positive controls and CuS (0.4 mg mL$^{-1}$) was used as a negative control. iRTX (1 µM) was used as a TRPV1 antagonist. Laser: 30 cycles. β-actin was used as a loading control. Data are shown as mean ± S.D. of three independent experiments, and analyzed by Student's $t$-test. *$P$ < 0.05 for CuS-TRPV1 vs. untreated group, #$P$ < 0.05 for Cap vs. untreated group, &$P$ < 0.05 for Rap vs. untreated group

CuS-TRPV1 still maintained high levels of membrane-targeting activity (Supplementary Fig. 15). Moreover, long-term incubation of CuS-TRPV1 did not cause obvious cytotoxicity (Supplementary Fig. 16). These results suggest that CuS-TRPV1 possesses good biocompatibility and its stability is sufficient for application in complex living systems. Next, we chose plaque-bearing ApoE$^{-/-}$ mice as an in vivo model. With the purpose of an accurate activation of TRPV1 signaling, we first applied high-frequency ultrasound (US) and PA imaging to observe cardiovascular selectivity of CuS-TRPV1 in ApoE$^{-/-}$ mice. CuS-TRPV1 NPs (10 mg kg$^{-1}$) were intravenously administered and US and PA images were acquired at different time-points post-injection. As shown in Fig. 5a, b, the PA contrast in the region of the aortic arch increased over time (1.81 ± 0.15 and 2.94 ± 0.39 at 1 and 2 h post-injection) compared to that before injection. The PA contrast decreased to 1.59 ± 0.47 due to the clearance of CuS-TRPV1 in vivo at 4 h post-injection, coinciding with ex vivo PA imaging results (Supplementary Fig. 17). Moreover, when CuS without TRPV1 was injected, only a slight increase in PA signal was observed due to the lack of targeting capability. It has been identified that blood vessels surrounding the atherosclerotic plaques are defective and immature[35–37]. Once a large proportion of CuS-TRPV1 accumulated in the aortic arch at 2 h post-injection, the leaky vasculature allowed them to penetrate and retain in plaques. To better confirm this, sequential sections of aortic arch, including both atherosclerotic and healthy aortas, were stained for α-smooth muscle actin (α-SMA) and immunoglobulin G (IgG), respectively. Compared to IgG background control, immunofluorescence micrographs of atherosclerotic aortas (Fig. 5c, d) revealed significant levels of CuS-TRPV1 (green fluorescence) that overlaid with the VSMC marker α-SMA (red fluorescence) at intimal and media layers. Lack of colocalization in healthy aortas (Supplementary Fig. 18) demonstrated that CuS-TRPV1 can easily pass through the atherosclerotic arterial wall and be trapped by plaque VSMCs within intimal and media layers. Without TRPV1-mediated targeting, non-coated CuS NPs did not show such specific localization (Fig. 5c, e). These results

together demonstrated the plaque-targeting and penetrating ability of CuS-TRPV1. The biodistribution and safety of CuS-TRPV1 were further confirmed by ex vivo organ inductively coupled plasma mass spectrometry (ICP-AES) for 2, 4, 24, and 72 h post-injection (Supplementary Fig. 19). By virtue of TRPV1-mediated targeting and enhanced permeation and retention (EPR) effect of atherosclerosis[38, 39], the accumulation of CuS-TRPV1 in aortic arch reached ~24.6% ID/g at 2 h. On the other hand, low accumulation in the heart, liver, spleen, lung and kidney reflected a relatively low uptake in non-target organs at this time-point. Similar to other CuS NPs-based theranostic agents[40, 41], the majority of intravenously injected CuS-TRPV1 were taken up by the reticuloendothelial system (RES) with liver uptake being ~20.4% ID/g at 4 h. Liver uptake gradually decreased to ~14.2% ID/g at 24 h, suggesting that small CuS NPs can escape macrophage sequestration in RES and might be renally eliminated from the body. The increased uptake in the kidneys at 24 h (~16.8 ID/g) further confirmed this. After 72 h, Cu content decreased in most of the tissues and organs. Therefore, to ensure therapeutic efficacy and safety, we chose a frequency twice a week to perform the injection of the NPs and 2 h time-point to perform the NIR treatment.

For PA image-guided therapy of atherosclerosis, ApoE$^{-/-}$ mice (male, 6–8 weeks) on a high-fat diet were randomly distributed into seven groups ($n = 16$): (a) control group; (b) Laser group; (c) CuS group; (d) CuS + Laser group; (e) CuS-TRPV1 group; (f) CuS-TRPV1 + Laser group; (g) capsaicin group. The 150 µL of CuS and CuS-TRPV1 PBS solution were administered intravenously (group c-f), whereas 150 µL of capsaicin PBS/ethanol solution was administered intragastrically (group g), both at a concentration of 10 mg kg$^{-1}$. The mice in group (a) and (b) received the same volume of PBS. Two hours post injection, group (b), (d), and (f) were irradiated using NIR laser (980 nm, 5 W cm$^{-2}$, spot diameter size 0.5 cm, Fig. 6a) at the cardiac region for 30 cycles twice weekly. After 12 weeks, the increased AMPK phosphorylation and LC3II/LC3I ratio indicated significant TRPV1 activation in mice treated with CuS-TRPV1 + Laser

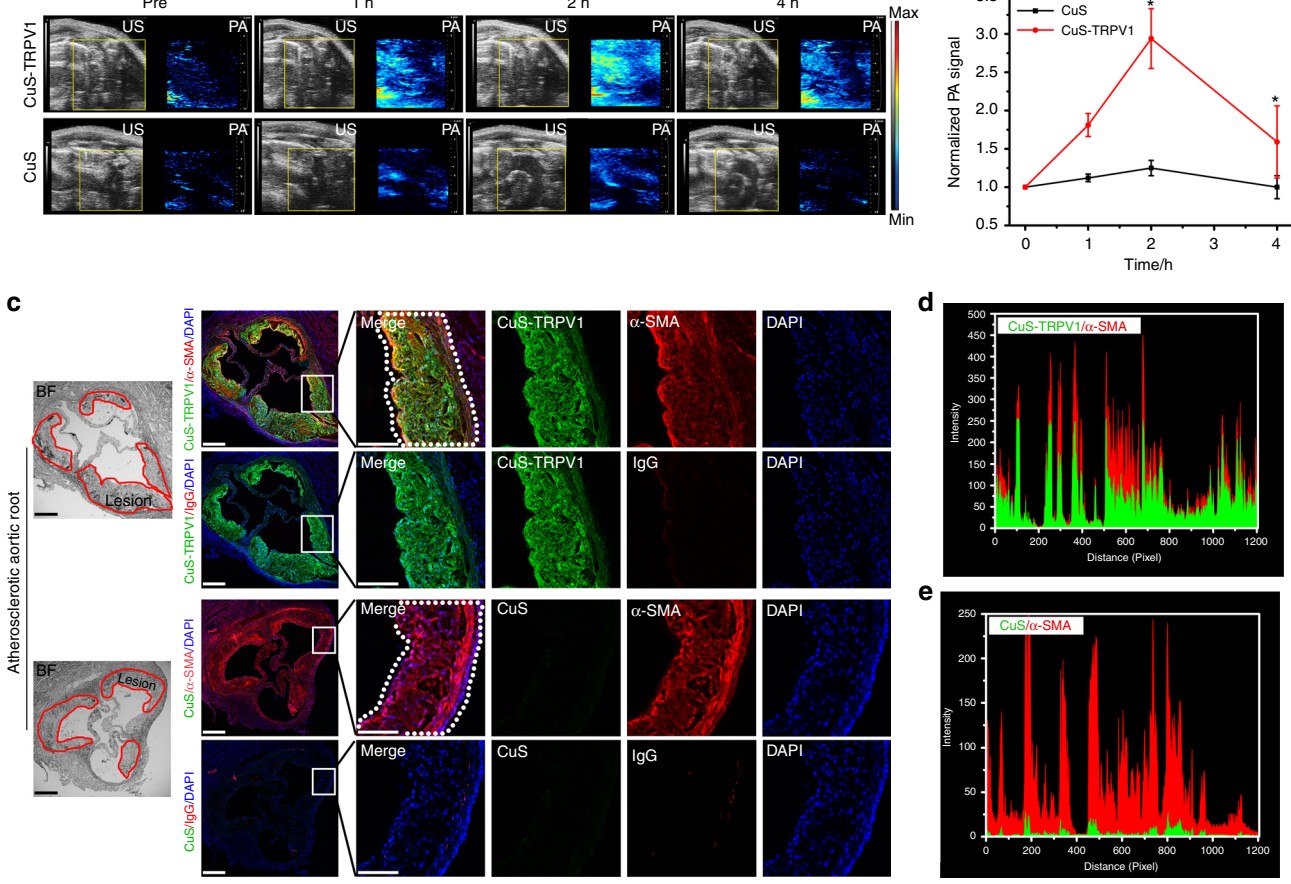

**Fig. 5** Localization of CuS-TRPV1 in high-fat-fed ApoE$^{-/-}$ mice. **a** US and PA imaging of the aortic arch in ApoE$^{-/-}$ mice intravenously received CuS-TRPV1 (upper panel) and CuS (lower panel). Yellow squares were used to indicate aortic arch location. **b** PA intensity of the aortic arch at different time-points. Data are shown as mean ± S.D. ($n = 3$), and analyzed by Student's $t$-test. *$P < 0.05$ vs. pre. **c** Representative immunofluorescent micrographs of aortic roots sections demonstrating CuS-TRPV1 within atherosclerotic plaque at 2 h post-injection. From right, blue (DAPI), red (VSMC maker α-SMA or background control IgG), green (fluorescein-conjugated CuS-TRPV1 or CuS) and merge. The red line indicated the lesion border. The white square gave a view of the magnification of a region of lesion as indicated on the right (Scale bar = 75 μm, $n = 3$ mice per group). **d** and **e** Fluorescence intensity profiles of the regions enclosed by the white dotted lines

(Supplementary Fig. 20), which led to subsequent ABCA1-mediated cholesterol efflux from aortic arches (Supplementary Fig. 21). As a result, CuS-TRPV1 + Laser treatment performed the best in targeted limiting atherosclerotic lesion progression in ApoE$^{-/-}$ mice compared to other controls. Oil red O staining (Fig. 6b–f and Supplementary Figs. 22, 23) displayed a significant 54.2% reduction ($167 × 1000$ μm$^2$ vs. $365 × 1000$ μm$^2$) in aortic root lesion areas and 72.3% reduction (3.1% vs. 11.2%) in en face prepared aortic arch lesion areas (within the dashed box above) compared with PBS group. In comparison, a smaller decrease in lesions of aortic root sections (31.2%, $251 × 1000$ μm$^2$ vs. $365 × 1000$ μm$^2$) and en face aortic arch preparations (49.1%, 5.7% vs. 11.2%) were observed in capsaicin group, despite having a significant therapeutic effect (27.5% reduction, 0.50% vs. 0.69%) in en face thoracic-abdominal aortic preparations (within the dashed box below). The main reason for this is that capsaicin is diffusible and lack specificity; therefore, it cannot confine its therapeutic effects to a specific area of body (cardiac region in this experiment) as seen in CuS-TRPV1 + Laser treatment. Besides the lack of specificity, toxicity should be considered in the long-term use of capsaicin. After 12 weeks, a slower weight increase (~5.8 g) and obvious liver damage (cellular shrinkage and steatosis in liver cells) were observed in capsaicin group (Supplementary Figs. 24, 25). In contrast, the mice in CuS-TRPV1 + Laser group continued

to grow without noticeable organ damage during the course of therapy. These results verified that CuS-TRPV1 was safe and well tolerated in mice and hold great promise for PA image-guided therapy of high-fat-induced atherosclerosis.

## Discussion

Atherosclerosis is an inflammatory disease that is initiated by lipid-mediated vascular inflammation of the vessel wall, which promotes the formation of VSMC/Mφ-derived foam cells. Strategies for developing therapeutics for this disease include the search for key regulators in VSMCs that promote autophagy and reduce lipid accumulation. We hypothesized that the thermo-sensitive cation channel TRPV1 may be a candidate based on previous work, which suggested that its activation by capsaicin impede VSMC foam cell formation through autophagy induction. However, capsaicin is unable to activate TRPV1 in a safe, accurate and controlled manner, limiting its clinical application and therapeutic outcome.

Our study demonstrated that TRPV1 signaling can be turned on in VSMCs and in ApoE$^{-/-}$ mice by using a CuS-TRPV1 photothermal switch. Our switch takes advantage of the strong NIR absorbance of CuS NPs, which allows PA imaging of cardiac vasculature with high-resolution and gating of thermosensitive

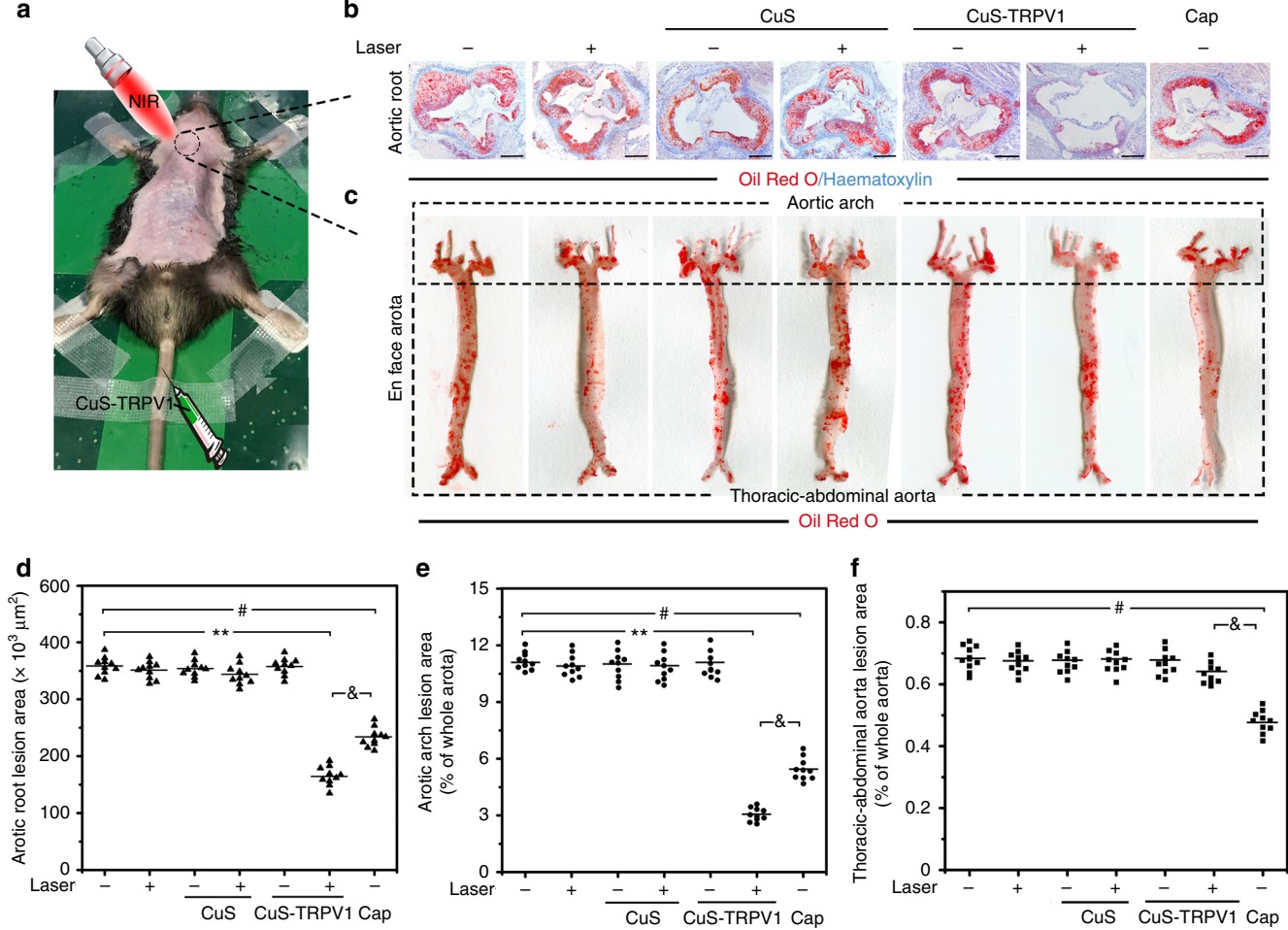

**Fig. 6** In vivo photothermal activation of TRPV1 reduces atherosclerotic lesions in ApoE$^{-/-}$ mice on a high-fat diet. **a** NIR laser treatment. **b** Representative images of Oil Red O-stained aortic root sections. Haematoxylin was used as a counterstain. Scale bar = 250 μm. **c** Representative images of Oil Red O-stained en face aortic preparations. Within the dashed box above is the aortic arch and below it is the thoracic-abdominal aorta. **d–f** Quantification of Oil Red O-stained area. Data are shown as mean ± S.D. ($n = 10$), and analyzed by Student's $t$-test. $^{**}P < 0.01$ for CuS-TRPV1 + Laser vs. PBS, $^{#}P < 0.05$ for Cap vs. PBS, $^{&}P < 0.05$ for CuS-TRPV1 + Laser vs. Cap

TRPV1 cation channel with high temporal and spatial precision. Activation of TRPV1 by CuS-TRPV1 with NIR irradiation rescued the autophagy impaired by oxLDL via activation of Ca$^{2+}$-AMPK signaling pathways and increased ABCA1-dependent cholesterol efflux. This ultimately led to the inhibition of VSMC foam cell formation and attenuation of atherosclerotic lesion in ApoE$^{-/-}$ mice. Because CuS-TRPV1 switch can confine its therapeutic effects to the cardiac region, a more effective outcome was obtained compared to TRPV1 agonists capsaicin. This serves an important purpose because it can target treatment of areas prone to atherosclerotic lesions, such as aortic arch, carotid artery and femoral artery, which cannot be achieved by capsaicin alone. More importantly, long-term activation of TRPV1 by our switch showed no obvious in vivo toxicity. In virtue of these advantages, the developed CuS-TRPV1 switch has the potential to be a powerful tool for accurate image-guided therapy of atherosclerosis.

## Methods

**Materials**. Copper(II) chloride (CuCl$_2$·2H$_2$O), sodium sulfide (Na$_2$S·9H$_2$O), Trisodium citrate (C$_6$H$_5$Na$_3$O$_7$·2H$_2$O), NaCl, Sodium dodecyl sulfate (SDS) were obtained from Sinopharm Chemical Reagent. Co. Ltd. (Shanghai, China). Capsaicin, rapamycin, 50-iodo-resiniferatoxin (iRTX), 1-ethyl-3-(3-dimethylaminopropyl) carbodiimide hydrochloride (EDC), Nhydroxysuccinimide (NHS), 5(6)-

Carboxyfluorescein N-hydroxysuccinimide ester (NHS-fluorescein), 3-(4,5-dimethyl-thiazol-2-yl)-2,5-diphenyltetrazolium bromide (MTT), 6-aminofluorescein were purchased from Sigma-Aldrich (St. Louis, MO, USA). 4% paraformaldehyde, Triton X-100 were from Solarbio (Beijing, China). Anti-TRPV1 (BA2589) and Antibody against ABCA1 (BA1541-2) were from Boster (Wuhan, China). Antibody against LC3B (AL221), mouse IgG (A7028), Fluo-3 AM, polyvinylidene fluoride (PVDF) membrane, RIPA lysis buffer, antibody against p-AMPKα (Thr 172) (AA393) and the enhanced chemiluminescence (ECL) substrate were obtained from Beyotime Biotechnology (Shanghai, China). Antibody against AMPK (ab3760) and β-actin (ab8227) were from Abcam (Burlingame, CA, USA). Antibody against α-SMA (55135-1-AP) was from SanYing Biotechnology (Wuhan, China). Peroxidase-conjugated affinipure goat anti-rabbit IgG (ZB-2301), fluorescein-conjugated affinipure goat anti-rabbit IgG (ZF-0311), Rhodamine (TRITC)-conjugated affinipure rabbit anti-mouse IgG (ZF-0313) were from ZSGB-Bio (Beijing, China). OxLDL was from DGCS. Biomart (Beijing, China). All chemicals and solvents used were of analytical grade. Water was purified with a Sartorius Arium 611 VF system (Sartorius AG, Germany) to a resistivity of 18.2 MΩ cm.

**Preparation of CuS and CuS-TRPV1**. For the synthesis of citrate-capped CuS NPs, 10 mL of CuCl$_2$·2H$_2$O (1.077 mg mL$^{-1}$) aqueous solution and 10 mL of sodium citrate (1.0 mg mL$^{-1}$) aqueous solution were added into 30 mL of water. The mixture was stirred for 30 min at room temperature. After that, 50 μL of Na$_2$S·9H$_2$O (743.92 mg mL$^{-1}$) aqueous solution was added to the mixture stirred for another 5 min before transferring to a 90 °C oil bath. The reaction was kept at this temperature for 15 min, forming green-colored CuS-Citrate NPs.

Then, CuS-TRPV1 was obtained by coupling the amino groups of TRPV1 antibody with the carboxyl groups on the surface of CuS-Citrate to form the amido

bonds[11]. 3.36 μmol of EDC and 3.36 μmol of NHS were added to 1 mL of CuS-Citrate (0.4 mg mL$^{-1}$) solution and reacted for 30 min at room temperature to activate carboxylate groups. Then 2 μg TRPV1 antibody was added to the solution under stirring for 12 h, followed by removing free antibody by filtration (MWCO = 1000 kDa). The prepared CuS-TRPV1 nanoconjugates were washed with deionized water for several times and redispersed in PBS (pH = 7.4) for further use.

**Characterization of CuS and CuS-TRPV1.** High resolution transmission electron microscopy (HRTEM) was carried out on a JEM-2100 electron microscope. Absorption spectra were measured on a U-4100 UV-visible-NIR spectro-photometer (HITACHI, Japan). Gel electrophoresis was run using the Mini-PROTEAN Tetra cell (Bio-Rad, Hercules, CA, USA). Fourier transform infrared (FT-IR) spectrum was collected on a Nicolet Impact 410 FTIR spectrometer in the range of 400–4000 cm$^{-1}$.

**Conjugation of fluorescein-labeled CuS-TRPV1 and CuS.** For conjugation of CuS-TRPV1 with fluorescein, CuS-TRPV1 (0.4 mg) was suspended in 1 mL PBS by ultrasonication for 5 min at room temperature. Then NHS-fluorescein (0.2 mg), which was dissolved in 10 μL of DMSO, was mixed with the CuS-TRPV1 solution for 1 h at room temperature. The final fluorescein-labeled nanoparticles were purified by centrifugation (11,000 g, 20 min, 4 °C) and washed with PBS/DMSO until no NHS-fluorescein was detected in the supernatant. For conjugation of CuS with fluorescein, CuS (0.4 mg) was suspended in 1 mL PBS (pH 7.4) by ultra-sonication for 5 min at room temperature. Afterward, 3.36 μmol of EDC and 3.36 μmol of NHS were added to CuS PBS solution and reacted for 30 min at room temperature to activate carboxylate groups. Then 10 μL of acetone solution containing 0.2 mg of 6-aminofluorescein was added and continuously stirred at room temperature overnight. The fluorescein-labeled particles were isolated via centrifugation (11,000 g, 20 min, 4 °C) and repeatedly washed with PBS/acetone until no 6-aminofluorescein was detected in the supernatant.

**Photothermal effect and photoacoustic signal of CuS-TRPV1.** To study the photothermal effect, CuS-TRPV1 with varied concentration (0.05 to 0.4 mg mL$^{-1}$) was suspended in PBS, and subjected to 980 nm laser with different laser power density (0.2 to 5.0 W cm$^{-2}$). The temperature at different time-points was monitored by FLIR S6 camera (FLIR Systems Inc., Wilsonville, OR). Repeated heat/cool cycles (30 s irradiation followed by a 30 s cooling period) were performed with the same instrument. The photoacoustic signals of CuS-TRPV1 with different concentration (0, 0.1, 0.2, 0.4 mg mL$^{-1}$) were recorded on Endra Nexus128 PA tomography system (Ann Arbor, Michigan).

**Cell culture.** Rat aortic vascular smooth muscle cells were obtained from Dingguo Changsheng Biotechnology Co., Ltd. (Beijing, China). Cells were cultured in Dulbecco's modified Eagles medium (DMEM) with 10% fetal bovine serum and 100 U/mL of 1% antibiotics penicillin/streptomycin and maintained at 37 °C in a 5% CO$_2$/95% air humidified incubator (MCO-15AC, SANYO). To ensure maintenance of the contractile phenotype, we used cells in the second to eighth passages for each experiment.

**Localization of CuS-TRPV1 in VSMCs.** Intracellular localization of CuS-TRPV1 was monitored by using TEM and IFC. For TEM measurements, VSMCs were incubated with sterilized CuS-TRPV1 and CuS (0.4 mg mL$^{-1}$) for 2 h, respectively. After that, cells were trypsinized, centrifuged, and fixed in 2.5% glutaraldehyde in sodium cacodylate buffer (pH 7.4) for 1 h at room temperature and rinsed. Cells were then post fixed 1 h in 2% osmium tetroxide with 3% potassium ferro-cyanide and rinsed, followed by en bloc staining with a 2% aqueous uranyl acetate solution and dehydration through a graded series of alcohol (50, 70, 80, 95, 100%). Then, cells were put into two changes of propylene oxide, a series of propylene/epon dilutions and embedded in 100% Epon. The 70 nm thin sections were cut on a Leica UC6 ultramicrotome, and images were taken on a JEOL 1200 EX (JEOL, Ltd. Tokyo, Japan) using an AMT 2k digital Camera. For IFC measurements, the fluorescein-conjugated CuS-TRPV1 and CuS was respectively cultured with VSMCs for 2 h at a concentration of 0.4 mg mL$^{-1}$. After the incubation time, stained cells were trypsinized, fixed with 4% formaldehyde in PBS and centrifuged (1000×g) to obtain a pellet of about 10$^6$ cells in 50 μL. Cell images were acquired using ImageStream$^X$ multispectral imaging flow cytometer, collecting 20,000 events per sample at 40 × magnification. A 488 nm wavelength laser was used to excite fluorescein. The fluorescence images were collected using the 500–560 nm spectral detection channels. For stained cells, unstained controls were used to compensate fluorescence between channel images on a pixel-bypixel basis. Cell images were analyzed using IDEAS® image-analysis software (Amnis).

**Binding affinity of CuS-TRPV1.** VSMCs were seeded at an initial density of 5 × 10$^4$ cells/dish in 20-mm glass bottom dishes and incubated for 24 h before adding the test substance. Then fluorescein-conjugated CuS-TRPV1 and CuS was respectively cultured with VSMCs for 2 h at a concentration of 0.4 mg mL$^{-1}$. In blocking group, excess amount of TRPV1 antibody (0.5 mg mL$^{-1}$) was added to cells 30 min before fluorescein-conjugated CuS-TRPV1. In NIR group, cells were

further irradiated by the 980 nm laser (5 W cm$^{-2}$) for 30 cycles. After the wash step with PBS, the binding affinity of CuS-TRPV1 for TRPV1 on VSMCs was examined by TCS SP8 confocal laser scanning microscopy (Leica Co., Ltd. Germany) with 488 nm excitation. After confirming the cell focusing ranges from the bottom to top, z-height images were captured at 0.5, 2, 3.5, and 5 μm from the bottom and analyzed by Leica Application Suite X (Leica Co., Ltd. Germany) with a ×63 objective lens.

**Determination of local temperature and cytotoxicity.** VSMCs were seeded in 96-well plates (10$^4$ cells/well) and cultured overnight. Then fresh medium containing sterilized CuS-TRPV1 (0.4 mg mL$^{-1}$) was added to each well and incubated for 2 h. One of the CuS-TRPV1 groups was irradiated with a 980 nm laser (5.0 W cm$^{-2}$) for one heat/cool cycle. The temperature changes of the culture medium were detected as mentioned above. The other CuS-TRPV1 groups were irradiated under the same conditions for multiple times (10, 20, and 30 cycles). Cell viability was evaluated by MTT assay according to the previously reported method[42] and measured in a microplate reader (RT 6000, Rayto, USA). Cells incubated with CuS-TRPV1 without NIR irradiation were served as a negative control.

**Photothermal activation of TRPV1 channels.** For confocal fluorescence imaging of Ca$^{2+}$ influx, VSMCs were seeded at an initial density of 1 × 10$^6$ cells/dish in 20-mm glass bottom dishes and cultured overnight. Then fresh medium containing sterilized CuS-TRPV1 (0.4 mg mL$^{-1}$) was added to each well and incubated for 2 h. Experimental groups were irradiated with 980 nm laser (5.0 W cm$^{-2}$) for one heat/cool cycle. The laser spot was adjusted to cover each well. After that, they were incubated with Ca$^{2+}$ indicator (Fluo-3 AM, 10 μM) for 30 min in darkness at 37 °C. Fluorescence images of the cells were examined by TCS SP5 confocal laser scanning microscopy (Leica Co., Ltd. Germany) through a ×40 objective lens at 488 nm excitation. For western blot analysis of AMPK phos-phorylation, VSMCs were seeded in 6-well cell plates (10$^6$ cells/well) and cultured overnight. Sterilized CuS-TRPV1 (0.4 mg mL$^{-1}$) was added to each well and incubated for 2 h. After irradiation by 980 nm laser (5.0 W cm$^{-2}$) for 10 heat/cool cycles, cells were collected, washed twice with cold PBS and centrifuged at 1200×g for 5 min. The cell pellet was resuspended in RIPA lysis buffer with 1 mM PSMF protease inhibitor (Beyotime, China). The amount of protein was measured using a protein assay kit (Beyotime, China). Equal amounts of protein (50 μg) were resolved on 10% SDS-PAGE gels and electro-blotted onto a polyvinylidene fluoride (PVDF) membrane and blocked with TBS containing 0.05% Tween-20 and 5% nonfat milk powder for 1 h. Next, membranes were incubated overnight with primary antibodies against p-AMPK (1:1000), AMPK (1:1000), and β-actin (1:1000), followed by HRP-conjugated secondary antibody (1:50,000) for 1 h. Detection was carried out by incubating membranes for 5 min with enhanced chemiluminescence reagent, followed by exposure to ChemiDoc™ Touch Imaging system (Bio-Rad, Hercules, CA, USA). Cells incubated with capsaicin (Cap, 1 μM) were used as a positive control and CuS (0.4 mg mL$^{-1}$) was used as a negative control. iRTX (1 μM) was used as a TRPV1 antagonist. The bands were quantified by densitometry using Image Lab 5.2 software (Bio-Rad). Uncropped blots are presented in Supplementary Figs 26, 27.

**Photothermal activation of autophagy.** For characterization of autophagosomes, VSMCs were seeded in 6-well cell plates (10$^6$ cells/well) and cultured overnight. After pretreatment with oxLDL (80 μg mL$^{-1}$) for 24 h, VSMCs were incubated with sterilized CuS-TRPV1 (0.4 mg mL$^{-1}$) for 2 h and irradiated by 980 nm laser (5.0 W cm$^{-2}$) for 30 heat/cool cycles. The accumulation of typical autophagosomes with double membranes was identified morphologically by TEM following the same procedure as described above. For western blot analysis of LC3II and LC3I expression, VSMCs were seeded in 6-well cell plates (10$^6$ cells/well) and cultured overnight. After pretreatment with oxLDL (80 μg mL$^{-1}$) for 24 h, VSMCs were incubated with sterilized CuS-TRPV1 (0.4 mg mL$^{-1}$) for 2 h and irradiated by 980 nm laser (5.0 W cm$^{-2}$) for 0, 10, 20, 30 heat/cool cycles. Cells incubated with capsaicin (1 μM) and rapamycin (10 nM) were used as positive controls, and cells incubated with CuS (0.4 mg mL$^{-1}$) were used as a negative control. iRTX (1 μM) was used as TRPV1 antagonist. Then, all groups of cells were collected and lysed. Changes in LC3II and LC3I expression were analyzed by western blot following the same procedure as described above (12% SDS-PAGE gels, primary antibody against LC3B: 1:1000). Uncropped blots are presented in Supplementary Fig. 26.

**Foam cell formation assay.** Foam cells were quantified by Oil Red O staining and intracellular total cholesterol content[34]. VSMCs were seeded at an initial density of 1 × 10$^6$ cells/dish in 20-mm glass bottom dishes and cultured overnight. After pretreatment with oxLDL (80 μg mL$^{-1}$) for 24 h, VSMCs were incubated with sterilized CuS-TRPV1 (0.4 mg mL$^{-1}$) for 2 h and irradiated by 980 nm laser (5.0 W cm$^{-2}$) for 30 heat/cool cycles. Cells incubated with capsaicin (Cap, 1 μM) were used as a positive control and CuS (0.4 mg mL$^{-1}$) was used as a negative control. After washing with PBS, VSMCs were fixed with 4% paraformaldehyde, washed with 60% isopropanol, and stained for lipid droplets with Oil Red O in 60% isopropanol. Foam cells were photographed under a fluorescence microscope with a ×40 objective lens (Leica DMI3000B, Germany). Intracellular total cholesterol content was determined by cholesterol-cholesteryl ester quantification assay kit (Abcam,

ab65359) according to manufacturer's protocol. In brief, lipids were extracted from cells with chloroform: isopropanol: NP-40 (7:11:0.1), pelleted and organic phase air dried. Assay was performed in the presence of cholesterol esterase with florescence measured on a microplate reader ($\lambda$ex = 535/$\lambda$em = 595). The cholesterol concentration = (A/V) × D (A = amount of cholesterol ($\mu$g) determined from Standard Curve, V = volume of sample ($\mu$L) added into the reaction well, D = Dilution Factor). Data was normalized to untreated group.

**Determination of ABCA1 mediated-cholesterol efflux.** For western blot analysis of ABCA1 expression, VSMCs were seeded in 6-well plates ($10^6$ cells/well) and cultured overnight. After pretreatment with oxLDL (80 $\mu$g mL$^{-1}$) for 24 h, VSMCs were incubated with sterilized CuS-TRPV1 (0.4 mg mL$^{-1}$) for 2 h and then irradiated by 980 nm laser (5.0 W cm$^{-2}$) for 30 heat/cool cycles. Cells incubated with capsaicin (1 $\mu$M) was used as a positive control, and cells incubated with CuS (0.4 mg mL$^{-1}$) was used as a negative control. Then, each group of cells were collected and lysed. Changes in ABCA1 expression were analyzed by western blot following the same procedure for AMPK (6% SDS-PAGE gels; primary antibody against ABCA1: 1:1000). Uncropped blots are presented in Supplementary Fig. 27. The effect of CuS-TRPV1 on cholesterol efflux in VSMCs was examined with the Cholesterol efflux assay kit (Sigma-Aldrich). VSMCs were seeded in 6-well plates ($10^6$ cells/well) and cultured overnight. After pretreatment with fluorescent-labeled cholesterol (5 $\mu$M) for 24 h, VSMCs were washed and incubated with sterilized CuS-TRPV1 (0.4 mg mL$^{-1}$) for 2 h, followed by 980 nm laser irradiation (5.0 W cm$^{-2}$) for 30 heat/cool cycles. Cells incubated with capsaicin (Cap, 1 $\mu$M) were used as a positive control and CuS (0.4 mg mL$^{-1}$) was used as a negative control. After an additional incubation for 6 h, the medium in each group was transferred to a fresh 96-well plate and the fluorescence was measured using a microplate reader (Fm, $\lambda$ex = 482/$\lambda$em = 515 nm). The cells in each group were lysed in a lysis buffer and the lysate was added to the transferred medium in another fresh 96-well plate. After mixing the contents of the wells, the fluorescence was measured (Fc, $\lambda$ex = 482/$\lambda$em = 515 nm). The percent Cholesterol efflux (C) was calculated as follows: C = 100% × Fm/Fc.

**Immunocytochemistry.** VSMCs were seeded on glass slides ($10^6$ cells) and cultured overnight. After pretreatment with oxLDL (80 $\mu$g mL$^{-1}$) for 24 h, VSMCs were incubated with sterilized CuS-TRPV1 (0.4 mg mL$^{-1}$) for 2 h and irradiated by 980 nm laser (5.0 W cm$^{-2}$, spot size 300 $\mu$m in diameter) for 30 heat/cool cycles. Then VSMCs were washed with PBS and fixed with 4% paraformaldehyde for 20 min at 4 °C. Nonspecific proteins were blocked with normal goat serum blocking solution for 1 h at 37 °C. Then primary antibody for LC3B was added and incubated overnight at 4 °C. After washing the cells with PBS for 3 times, FITC-labeled secondary antibody was added and incubated in the dark for another 1 h at 37 °C. Finally, the nucleus was counterstained with DAPI (10 mg mL$^{-1}$) for 5 min, washed with PBS and placed into serial diluted glycerol solution from 100% to 50% for confocal microscopy observation (TCS SP8 with a ×10 objective lens, Leica Co., Ltd. Germany).

**Localization of CuS-TRPV1 in plaque-bearing ApoE$^-$/$^-$ mice.** ApoE$^-$/$^-$ mice were purchased from Changzhou Cavens Laboratory Animal Co. Ltd. (Changzhou, China) and were fed ad libitum and housed under controlled lighting conditions (12 light:12 dark). They were maintained under specific pathogen-free conditions. All animal care and experimental protocols complied with the Animal Management Rules of the Ministry of Health of the People's Republic of China and were approved by the Animal Care Committee of Shandong Normal University (Approved Number: AEECSDNU2015005) authorized by Shandong Experimental Animal Center. At 6–8 weeks of age, mice (male, ~20 g) were fed with a high-fat diet (HFD, 20% fat, 20% sugar, and 1.25% cholesterol) for 12 weeks. After plaque builds up within arteries, 150 $\mu$L CuS-TRPV1 or CuS (fluorescein-conjugated CuS-TRPV1 or CuS) at the concentration of (10 mg kg$^{-1}$) was intravenously injected. Localization of the particles in mice was monitored by using in vivo US and PA imaging, and ex vivo PA imaging and histology study. First, plaque-bearing mice (n = 3) were anesthetized using 1.5% isoflurane with medical air at a flow of 2 L min$^{-1}$. Hair was removed over areas of interest using a depilatory cream. For aortic arch imaging, the chest area of mice was gently inflated with US gel. US and PA imaging were performed on a Vevo® LAZR Imaging Systerm (VisualSonics Inc. New York, NY), before and after an intravenous injection of 150 $\mu$L CuS or CuS-TRPV1 at the concentration of (10 mg kg$^{-1}$), respectively. Anatomical features of the aortic arch were collected using B-mode at frequency of 30 MHz, power of 100%, and 2D gain of 27 dB. PA signals of CuS-TRPV1 and CuS were collected using Nanostepper-mode at 915 nm wavelength, frequency of 30 MHz, power of 100% and PA gain of 40 dB (LZ400 transducer, VisualSonics, Canada). Within a region of interest (ROI), PA intensities were quantified with VevoCQ™ Software (VisualSonics, Canada) and normalized to the pre-injection levels. Second, ex vivo PA imaging was further carried out to confirm that PA signal intensities based on in vivo imaging truly represented the CuS-TRPV1 distribution in ApoE$^-$/$^-$ mice. At 1, 2, 4, and 6 h post-injection of CuS-TRPV1, aortic arches and major organs including heart, liver, spleen, lung and kidneys from ApoE$^-$/$^-$ mice (n = 3) were collected and visualized with Endra Nexus128. The PA signals in each tissue at indicated time-points were normalized to the corresponding pre-injection baseline

values. Third, at 2 h post-injection of fluorescein-conjugated CuS-TRPV1 or CuS, immunohistochemistry on frozen aortic root sections (n = 3) were performed to indicate where the particles reside within aortic arch. Healthy ApoE$^-$/$^-$ mice (n = 3) injected with fluorescein-conjugated CuS-TRPV1 were used as an additional control. Frozen sections of 5 $\mu$m were fixed in ice-cold acetone for 5 min and then blocked with normal goat serum. Sections were labeled with anti-mouse $\alpha$-SMA monoclonal antibodies overnight, followed by Rhodamine (TRITC)-conjugated secondary antibody for 1 h. Mouse IgG was used as a background control. The stained sections were mounted with DAPI-containing mounting medium and then viewed using a TCS SP8 confocal laser scanning microscopy (Leica Co., Ltd. Germany) with a ×10 objective lens.

**ICP-AES analysis.** Plaque-bearing ApoE$^-$/$^-$ mice (male, fed with high-fat diet for 12 weeks, n = 3) were intravenously injected with 150 $\mu$L of CuS-TRPV1 PBS solution (10 mg kg$^{-1}$). At 2, 4, 24, and 72 h postinjection, aortic arch and major organs were harvested immediately and weighed. Then these tissues were placed into borosilicate tube and digested in 70% w/v nitric acid containing 10 ppm Yttrium at 80 °C for 4 h. After cooled to room temperature, condensates were collected by centrifugation, and diluted to 5 mL with water. Samples were then filtered through a 0.45 $\mu$m hydrophilic PVDF membrane into 15 mL Falcon tubes. Standards were prepared by diluting 1000 p.p.m. cooper standards solution to 100 p.p.m., 10 p.p.m., 1 p.p.m., 0.2 p.p.m., 0.04 p.p.m. with 15% w/v nitric acid. ICP-AES was performed on the samples using the Optima 7300 (Perkin Elmer).

**In vivo anti-atherosclerotic efficacy.** To induce the development of atherosclerotic lesions, a total of 112 ApoE$^-$/$^-$ mice (6–8 weeks old, male, ~20 g) were fed with the high-fat diet (HFD, 20% fat, 20% sugar, and 1.25% cholesterol) and randomly divided into seven groups (n = 16): (a) control group; (b) Laser group; (c) CuS group; (d) CuS + Laser group; (e) CuS-TRPV1 group; (f) CuS-TRPV1 +Laser group; (g) capsaicin group. Then 150 $\mu$L PBS dispersions of CuS and CuS-TRPV1 were administered intravenously (group c–f), whereas 150 $\mu$L of capsaicin (40 mg mL$^{-1}$ ethanol stock solution diluted with PBS) was administered intragastrically (group g), both at a concentration of 10 mg kg$^{-1}$. The mice in group (a) and (b) received the same volume of PBS. Two hours post injection, group (b), (d), and (f) were irradiated using NIR laser (980 nm, 5 W cm$^{-2}$, spot diameter size 0.5 cm, Fig. 6a) at the cardiac region for 30 cycles. The other groups were not irradiated. This treatment was performed twice weekly and body weights of mice were recorded once weekly. After 12 weeks, all the mice were euthanized and perfused with PBS. The peripheral fat and connective tissue was removed as much as possible. To evaluate in vivo photothermal activation of TRPV1, aortic arches from each group mice (n = 6) were homogenized on ice in lysis buffer, respectively. Changes in p-AMPK/AMPK, LC3II/LC3I ratio and ABCA1 expression were analyzed by western blot following the same procedure as described above (primary antibody p-AMPK and AMPK: 1:1200; LC3B and ABCA1: 1:1000). Uncropped blots are presented in Supplementary Fig. 27. Total cholesterol levels were determined by cholesterol-cholesteryl ester quantification assay kit (Abcam, ab65359) according to manufacturer's protocol. Results were then normalized to the weight of the sample. To evaluate the extent of atherosclerotic lesion, two approaches were used. For Histological analysis[43], the upper sections of hearts from each group mice (n = 10) were embedded in paraffin, and 5-$\mu$m-thick serial sections were prepared. Sequential 5 $\mu$m thick cross-sections were cut until the appearance of the aortic sinus, which was recognized on unstained sections by the presence of the three leaflets of aortic valve and the bulging shape of the aorta. From this point on, every other section was collected on slides until the valve cusps were no longer visible and the aorta was uniformly round. Every 10th section was stained with Oil red O and haematoxylin. Photographs were digitized by using NIS-Elements imaging software (Nikon, Japan). Data were expressed as lesion area (×1000 $\mu$m$^2$). For en face analysis[33], the whole aortas from each group mice (n = 10) were opened longitudinally and stained with Oil red O. Photographs of the stained specimens were digitized for data analysis. The luminal lesion surface area was quantified by using NIS-Elements imaging software (Nikon, Japan). Data were expressed as the percentage of the aorta with positive Oil red O staining. For in vivo toxicity study, major organs of mice from each group were collected and sliced for haematoxylin and eosin (H&E) staining. Images were captured using a Leica DMI3000 microscope with a ×20 objective lens (Leica Microsystems, Wetzlar, GmbH).

**Statistical analysis.** Each experiment was repeated three times in duplicate unless stated otherwise. Data were presented as mean ± S.D. Comparisons between groups were analyzed using Student's t-test, P < 0.05 was considered statistically significant.

**Data availability.** All relevant data that support the findings of this study are available from the corresponding author upon reasonable request.

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

## Acknowledgements

This work was supported by National Natural Science Foundation of China (21535004, 21390411, 21775092, 21575082).

## Author contributions

W.G. and B.T. conceived and designed the experiments. W.G., Y.S. and Y.Z. performed the experiments. W.G., Y.S., W.C. and B.T. analyzed the data. Z.L., L.T. and G.C. contributed the schematic materials. W.G. and B.T. co-wrote the paper. W.G., Y.S. and M.C. edited the manuscript.

## Additional information

**Competing interests:** The authors declare no competing financial interests.

