## [Peer Review File · Nature Communications]

Reviewers' comments:

Reviewer #1 Remarks to the Author):

The manuscript by Wen Gao et al. entitled, "CuS nanoparticle as a photothermal switch for the control of TRPV1 signaling to attenuate atherosclerosis", describes remote controlled nanotherapeutics capable of selective stimulation of transient receptor potential vanilloid subfamily 1 (TRPV1) located on the plasma membrane of vascular smooth muscle cell (VSMC) in atherosclerosis. Near infrared (NIR) absorbing CuS-TRPV1 was designed to provide photoacoustic (PA) image-guided therapy as well as to stimulate thermosensitive TRPV1 in a controlled manner around the atherosclerotic plaques for autophagy activation and subsequent reduction in lipid accumulation. Remote regulation of cellular activity using inorganic nanoparticles is a potentially important technique for elucidating underlying physiological processes with promising therapeutic potential (Nature medicine 21; 92-98, 2015). A recent report on wireless magnetothermal deep brain stimulation shows remote neural excitation through magnetic nanoparticle-mediated TRPV1 activation (Science 347; 1477-1480, 2015). Despite the potential impact of this manuscript, it currently lacks sufficient description and results to support the overall strategy. The following major and minor comments should be addressed prior to consideration for publication in Nature Communications:

Major comments

1. Please demonstrate the binding affinity of CuS-TRPV1 for TRPV1 on VSMC. Supplementary Fig. 2 and Fig. 3 do not show that the TRPV1 antibodies conjugated on the surface of the CuS nanoparticles have biological activity and affinity. Also, please demonstrate the binding affinity of CuS-TRPV1 during repeated laser irradiations (0-30 cycles).
2. Please discuss the possibility of receptor desensitization resulting from repeated stimulation of TRPV1.
3. In Fig. 3a, 1) Please provide a TEM image of the whole cell and magnified images showing the plasma membrane, endosome, and lysosome to demonstrate the high density of nanoparticles confined to the plasma membrane. Why does CuS-TRPV1 remain on the surface membrane rather than undergo cell internalization via receptor-mediated endocytosis? 2) In the IFC image, please compare the fluorescence intensity of the CuS-TRPV1 with that of CuS and quantify. Please also include their statistical significance.
4. In Fig. 3c, 1) please include the p-AMPK data of CuS and CuS plus laser as negative control of the targeted nanoparticles. 2) Please show the p-AMPK amounts of Control, CuS-TRPV1, CuS-TRPV1 plus laser, and Capsaicin in TRPV1 inhibited VSMC to demonstrate that the p-AMPK increase roots from TRPV1. Please quantify the western blot analysis and include their statistical significance.
5. In Fig. 4b, please include the data for Untreated, CuS without laser, and CuS with laser (0-30 cycles). Please quantify all of the western blot analysis and include their statistical significance.
6. In Fig. 4c, please include CuS with laser and iRTX. Please also quantify the western blot analysis and include their statistical significance.
7. In Fig. S8, please include CuS and CuS+laser groups to prove the targeting ability of CuS-TRPV1 in lipid reduction.
8. Please compare Fig. S11 with CuS and quantify their membrane fluorescence intensities.
9. In Fig. 5, why is the CuS-TRPV1 cleared to an initial level at 4 h post-injection rather than

remained longer by TRPV1 binding?

10. In Fig. 6, 1) Please add CuS and CuS+laser groups to prove CuS-TRPV1's targeting ability in the therapy; 2) Please provide the western blot analysis and quantification data for pAMPK, LC3I, and LC3II of the aortic arch lesions.

Minor comments

1. References 10 - 12 are about magnetic nanoparticles. Please clarify terminology.
2. Please include 'synthesis and purification of CuS-TRPV1 and fluorescein conjugated CuS-TRPV1' in the supplementary methods section.
3. Please include a detailed description of the 'in vivo photothermal activation study' in the supplementary methods section (eg. number of mice, age, diet detail, therapeutic dose, administration route, injection frequency, etc.).
4. Please include a detailed description for the H&E staining of aortic root and the ORO staining in the supplementary methods section.
5. In Fig. S10, the stability of the CuS-TRPV1 should be evaluated in serum containing media at 37 °C.

Reviewer #2 Remarks to the Author:

This work has provided an alternative way to activate TRPV1 in vascular tissue from ApoE knockout mice using NIR-induced heating via CuS nanoparticle-conjugated TRPV1 antibodies. Major finding is that this novel technique could attenuate atherosclerosis. The results sounds interesting and the manuscript is well-written. However, there are still critical issues should be clarified to ensure the safety and reliability of this method.

Major points

1. VSMCs were incubated with CuS nanoparticle in in vitro experiments. But in vivo, endothelium is the first barrier of arterial wall in response to the stimuli from the blood stream. Why did the authors choose VSMCs as the in vitro cellular model rather than endothelial cells? Is there any specificity of CuS nanoparticles binding to VSMCs in vivo? The expression of TRPV1 is also abundant in endothelial cells (Yang DC, Cell metab, 2010; Xiong SQ, et al., Hypertension, 2016). How did authors exclude the role of endothelial TRPV1 activation in ameliorating atherosclerosis? How did CuS nanoparticle penetrate the plaque containing VSMCs in vivo? Authors should provide some experimental evidences for these key points.
2. The authors demonstrated that a near-infrared (NIR) light (5 W/cm²) could activate TRPV1 and reduced foam cell formation in vitro and attenuated atherosclerosis in vivo. It is well-known that temperature greater than 43 {degree sign}C can be able to activate TRPV1. TRPV1 is also expressed in myocardial, adipose and endothelial cells. How did the authors determine that the same power of NIR laser (5 W/cm²) could be efficient and accurate to activate the vascular TRPV1 in deep tissues, not involved in TRPV1 in other tissues?
3. In study in vivo, mice in the CuS-TRPV1-NIR group was irradiated at the cardiac region with NIR laser (980 nm, 5 W/cm², spot diameter size 0.5 cm). But Figure 6d showed that Enface aortic lesion of those mice was attenuated in the whole aorta, including abdominal aorta. Obviously, it was beyond the irradiated region. How did authors explain this outcome?
4. The results of Supplementary Fig.13 indicated that administration of capsaicin via intravenous injection caused an obvious weight loss. The authors thought it was the high toxicity of capsaicin. In fact, Zhang et al., reported that the activation of TRPV1 by dietary capsaicin could prevent

adipogenesis and obesity safely in mice (Zhang et al., *Cir Res*, 2007). A lot of human study and clinical trials also have shown that spicy food containing capsaicin can prevent obesity. Meanwhile, these differences may be related to the different drug-delivery way. Furthermore, capsaicin is rarely dissolved in water or PBS thus it is usually added in diet to treat animals. I wonder how it was dissolved in PBS and injected intravenously as a positive control in this work.

5. The plague area in the sections of aortic sinus should be quantified by Oil-red staining. And the significance of the statistical analysis on enface aortic lesions should be labeled.

6. Why did the authors choose a frequency twice a week to perform the injection of the nanoparticles and NIR treatment? Could the particles be catalyzed or removed from the circulatory system?

7. The statistics should be defined in the corresponding figure legends. The diagrams and statistical results should be presented (figure 4b, c). Specify the number of performed experiments in each figure legends. Use symbols (*, #) to show statistical differences in diagrams (figure 6e).

Reviewer #3 Remarks to the Author:

This paper by Gao et al., describes a nanoparticle approach for treating atherosclerosis using light-activated nanoparticles, to induce a heating effect that activates the TRPV1 channel. The approach is elegant, and this paper addresses a major unmet therapeutic need.

Several issues must be addressed to strengthen the data presented in this manuscript in order to be considered for publication in *Nat Comm*.

First, the in vivo data raises some major concerns. Specifically, 1) the authors indicate they performed a single intravenous injection of the targeted nanoparticles. The biodistribution of the particles to the artery must be presented clearly (Figure 5), it is very hard to determine from the images how effective the targeting is. Adding cross-sectional histology of the blood vessel that will indicate where the particles reside, relative the vessel and to the plaque is needed. This will indicate whether the targeted ligand is accessible on the plaque-coated lumen. Furthermore, 2) it seems that one treatment may not be sufficient in order to induce the dramatic therapeutic effect presented 12 weeks post treatment. Some kinetics of the plaque reduction would be reassuring. 3) In addition, one may question the need for a NIR group without nanoparticles to test whether the NIR alone can heat the cells.

Regarding the in vitro images. (Figure 3) Confocal z-height images of the nanoparticles on the cells will help better understand if the particles are internalized or on the cell surface.

Is the activation of the TRVP channel alone capable of reversing atherosclerosis or should other pathways be considered as possible contributors to this dramatic therapeutic effect.

In summary, this paper needs stronger evidence to support the conclusions.

Minor remarks:

1) Line 182 should read 'chose' (Next we choose high-fat-fed)

2) Figure S8b, y-axis should read 'cholesterol'

Reviewer #4 Remarks to the Author:

In this study entitled "CuS Nanoparticle as a photothermal switch for the control of TRPV1 signalling to attenuate atherosclerosis" Gao and colleagues develop a method that allow them to activate TRPV1 channels in the artery wall by using a near-infrared (NIR) red to activate TRPV1 channels and attenuate atherosclerosis in mice. Overall, the concept is of interest.

However, the results do not support author's conclusions and the characterization of plaque lesions is poorly done. It is recommended that an expert in atherosclerotic studies supervise this study.

There are numerous major weaknesses in this study including:

- 1) There is not quantification of plaque size in the aortic root (Fig. 6c). Why the authors overlook highlighting the biggest plaque in the CuS-TRPV1+laser group???
- 2) If activation of laser reduces lipid deposition, the authors should assess neutral lipid deposition (Oil Red O staining) in the cross-sectional analysis (aortic root).
- 3) How the laser is applied, it looks that influence lipid deposition in the whole aorta (enface plaque analysis).
- 4) The authors should also demonstrate that transient activation of TRPV1 influences p-AMPK and autophagy in vivo.

In summary, the data are preliminary and do not support author's conclusions.

Point-by-point response to the reviewer's comments

Response to reviewer 1:

Reviewer 1's comment 1-1:

The manuscript by Wen Gao et al. entitled, "CuS nanoparticle as a photothermal switch for the control of TRPV1 signaling to attenuate atherosclerosis", describes remote controlled nanotherapeutics capable of selective stimulation of transient receptor potential vanilloid subfamily 1 (TRPV1) located on the plasma membrane of vascular smooth muscle cell (VSMC) in atherosclerosis. Near infrared (NIR) absorbing CuS-TRPV1 was designed to provide photoacoustic (PA) image-guided therapy as well as to stimulate thermosensitive TRPV1 in a controlled manner around the atherosclerotic plaques for autophagy activation and subsequent reduction in lipid accumulation. Remote regulation of cellular activity using inorganic nanoparticles is a potentially important technique for elucidating underlying physiological processes with promising therapeutic potential (Nature medicine 21; 92-98, 2015). A recent report on wireless magnetothermal deep brain stimulation shows remote neural excitation through magnetic nanoparticle-mediated TRPV1 activation (Science 347; 1477-1480, 2015). Despite the potential impact of this manuscript, it currently lacks sufficient description and results to support the overall strategy. The following major and minor comments should be addressed prior to consideration for publication in Nature Communications.

Response 1-1:

We appreciate very much the reviewer's encouraging and constructive comments. Following the reviewer's advice, we have performed supplementary experiments to substantiate our novel conclusions. Point-by-point responses to reviewer's comments are showed as follows.

Comments 1-2:

Please demonstrate the binding affinity of CuS-TRPV1 for TRPV1 on VSMC. Supplementary Fig. 2 and Fig. 3 do not show that the TRPV1 antibodies conjugated on the surface of the CuS nanoparticles have biological activity and affinity. Also, please demonstrate the binding affinity of CuS-TRPV1 during repeated laser irradiations (0-30 cycles).

Response 1-2:

Following the reviewer's suggestion, a competitive binding study was developed based on confocal z-height imaging to evaluate the binding affinity of CuS-TRPV1 for TRPV1. TRPV1 channels on

VSMCs were first blocked by excess amount of TRPV1 antibody (0.5 mg/mL) and fluorescein-conjugated CuS-TRPV1 was subsequently incubated with cells. After confirming the cell focusing ranges from the bottom to top, confocal z-height images were captured at 0.5 μm , 2 μm , 3.5 μm and 5 μm from the bottom. Without TRPV1 blocking, CuS-TRPV1 bound with TRPV1 on VSMCs within 2 h, showing strong green fluorescence at focal plane near the surface (5 μm from the bottom) even after 30 cycles of NIR irradiation. In contrast, with TRPV1 blocking, only minimal fluorescent signal was observed when we focused at the surface of these cells. These results suggest that TRPV1 antibodies conjugated on the surface of the CuS nanoparticles retained their binding activity as free TRPV1 antibodies, which were capable of specifically targeting to the TRPV1 of VSMCs membrane (Figure R1 below and Figure S6 in the revised supporting information). Thus, the conjugated procedure and laser irradiation did not impair the binding ability of TRPV1 antibodies, as also demonstrated by others (*ACS Nano.*, 2015, 9(4): 3926-3934; *Nature Nanotech.*, 2016, 11: 525-532).

Figure R1. Representative confocal z-height images of VSMCs (a) incubated with fluorescein conjugated CuS-TRPV1 (0.4 mg/mL) for 2 h, (b) pretreated with TRPV1 antibody (0.5 mg/mL) and then incubated with fluorescein conjugated CuS-TRPV1 (0.4 mg/mL) for 2 h, and (c) incubated with fluorescein conjugated CuS-TRPV1 (0.4 mg/mL) for 2 h and then irradiated by the 980 nm laser (5 W/cm²) for 30 cycle. Scale bar = 5 μm .

Comments 1-3:

Please discuss the possibility of receptor desensitization resulting from repeated stimulation of TRPV1.

Response 1-3:

TRPV1 is a nonselective cation channel, highly permeable to Ca^{2+} , which can be activated by capsaicin and heat. Chen, R., *et al.* (*Science*, 2015, 347(6229): 1477-1480) have demonstrated remote neural excitation through the activation of the heat-sensitive TRPV1 by magnetic nanoparticles. To avoid prolonged exposure to noxious heat, 10 s pulses of alternating magnetic field at 15 kA/m and 500 kHz were delivered with 50 s rest intervals in between over the course of 20 min. Temperature gradients were sufficient to reach the TRPV1 activation threshold (43°C), thus successfully triggering Ca^{2+} influx and activating a subpopulation of neurons in deep brain tissue in mice. Similarly, Stanley, S.A., *et al.* (*Nature Med.*, 2015, 21, 92-98) have reported the utility of endogenously expressed nanoparticles *in vivo* in transducing both radio waves and magnetic fields for non-invasive control of TRPV1 and Ca^{2+} -dependent transgene expression. Repeated RF treatment (1 h of RF once a week during weeks 2-6 after virus injection) significantly decreases blood glucose in mice with hepatic expression of $\alpha\text{GFP-TRPV1/GFP-ferritin}$. These studies validated that repeated stimulation of TRPV1 by heat enables remote, robust and repeated temporal control of cell behaviors *in vitro and in vivo* rather than desensitization. Consistent with these results, our data also showed that LC3II/LC3I ratio (autophagy marker) was persistently increased over 30 irradiation cycles. Therefore, repeated laser irradiations of CuS-TRPV1 did not result in TRPV1 desensitization.

Comments 1-4:

In Fig. 3a, 1) Please provide a TEM image of the whole cell and magnified images showing the plasma membrane, endosome, and lysosome to demonstrate the high density of nanoparticles confined to the plasma membrane. Why does CuS-TRPV1 remain on the surface membrane rather than undergo cell internalization via receptor-mediated endocytosis? 2) In the IFC image, please compare the fluorescence intensity of the CuS-TRPV1 with that of CuS and quantify. Please also include their statistical significance.

Response 1-4:

Following the reviewer's suggestion, TEM image of the whole cell was presented in Figure R2 (Fig. 3a in the revised manuscript). It is obvious that, as compared with endosome, or lysosome, a high density of nanoparticles was confined to the plasma membrane after 2 h incubation. Because TRPV1 are

highly expressed on the plasma membranes of VSMCs and CuS-TRPV1 have the ability to bind TRPV1 through a specific monoclonal antibody interaction, a significant internalization was observed until 24 h after incubation (The loss in membrane intensity shown in Figure S14 in the revised supporting information).

Figure R2. Representative TEM images of VSMCs incubated with CuS-TRPV1 or CuS (0.4 mg/mL) for 2 h. Red arrows indicated localization of nanoparticles in (i) whole cell section under low magnification, (ii) partially magnified cell membrane and (iii) magnified endosome or lysosome. Scale bar = 1 μm .

IFC images of the fluorescently labelled CuS nanoparticles were shown in Figure R3 below (Fig. 3b in the revised manuscript). Compared with that of CuS-TRPV1, non-coated CuS nanoparticles did not show such a specific localization on VSMCs membrane, as they were internalized into the interior of cells. After 2 h of incubation, the membrane intensity was 6,520.78, 23% that of CuS-TRPV1.

Figure R3. Representative IFC images of VSMCs incubated with fluorescein-conjugated CuS-TRPV1 or CuS (0.4 mg/mL) for 2 h. Green fluorescence in the blue area indicated nanoparticle targeting to the cell membrane. Mean membrane intensity of CuS (blue area: 6,520.78) was 23% that of CuS-TRPV1 (blue area: 28,408.02). Data are present as mean \pm S.D. of three independent experiments, and analyzed by Student's *t*-test. *** $P < 0.001$.

Comments 1-5:

In Fig. 3c, 1) please include the p-AMPK data of CuS and CuS plus laser as negative control of the targeted nanoparticles. 2) Please show the p-AMPK amounts of Control, CuS-TRPV1, CuS-TRPV1 plus laser, and Capsaicin in TRPV1 inhibited VSMC to demonstrate that the p-AMPK increase roots from TRPV1. Please quantify the western blot analysis and include their statistical significance.

Response 1-5:

Following the reviewer's suggestion, we studied the phosphorylation levels of AMPK of in CuS and CuS plus laser group. As shown in Figure R4 (Fig. 3d in the revised manuscript), non-coated CuS NPs with and without laser irradiation exerted no effect on the expression of p-AMPK in VSMCs. Next, we further evaluated the involvement of TRPV1 in AMPK activation by using the TRPV1 antagonist, 50-iodo-resiniferatoxin (iRTX). As shown in Figure R5 (Figure S9 in the revised supporting information), iRTX significantly inhibited the TRPV1-activated AMPK in CuS-TRPV1-heated VSMCs, demonstrating that the increase in p-AMPK rooted from TRPV1. The western blots have been quantified and their significance has been confirmed by student's *t*-test.

Figure R4. Western blot analysis of the phosphorylation of AMPK in VSMCs induced by CuS-TRPV1 heating-evoked Ca^{2+} influx. VSMCs were incubated with 0.4 mg/mL CuS-TRPV1 and irradiated by the 980 nm laser (5 W/cm^2) for 10 cycles. Capsaicin (Cap, $1 \mu\text{M}$) was used as a positive control and CuS (0.4 mg/mL) was used as a negative control. The AMPK phosphorylation was normalized using β -actin. Data are shown as mean \pm S.D. of three independent experiments, and analyzed by Student's *t*-test. ** $P < 0.01$ for CuS-TRPV1 + laser vs. untreated group. # $P < 0.05$ for Cap vs. untreated group.

Figure R5. Western blot analysis of the involvement of TRPV1 in AMPK phosphorylation. VSMCs were incubated with CuS-TRPV1 (0.4 mg/mL) combined with iRTX (1 μ M) and irradiated by the 980 nm laser (5 W/cm²) for 10 cycles. Capsaicin (Cap, 1 μ M) was used as a positive control and CuS (0.4 mg/mL) was used as a negative control. The AMPK phosphorylation was normalized using β -actin. Data are shown as mean \pm S.D. of three independent experiments, and analyzed by Student's *t*-test. No statistical significance was detected.

Comments 1-6:

In Fig. 4b, please include the data for Untreated, CuS without laser, and CuS with laser (0-30 cycles). Please quantify all of the western blot analysis and include their statistical significance.

Response 1-6:

Following the reviewer's suggestion, we have determined the changes of LC3II/LC3I ratio in untreated, CuS with laser (0-30 cycles) groups. Western blotting data (Figure R6) showed no significant increase in these control groups. These corresponding results have been added in the supporting information of the revised paper (Figure S10 in the revised supporting information).

Figure R6. Western blot analysis of LC3II/LC3I ratio at the indicated irradiation cycle. VSMCs were incubated with 0.4 mg/mL CuS and irradiated by the 980 nm laser (5 W/cm^2) for different cycles (0, 10, 20 or 30 cycles). Data are shown as mean \pm S.D. of three independent experiments, and analyzed by Student's *t*-test. No statistical significance was detected.

Comments 1-7:

In Fig. 4c, please include CuS with laser and iRTX. Please also quantify the western blot analysis and include their statistical significance.

Response 1-7:

Following the reviewer's suggestion, we have added the western blot analysis of LC3II/LC3I ratio in CuS with laser and iRTX group. As shown in Figure R7 (Fig. 4c in the revised manuscript), iRTX exerted no effect on LC3II/LC3I ratio in CuS with laser irradiation groups.

Figure R7. Western blot analysis of LC3II/LC3I ratio under different conditions. Capsaicin (Cap, 1 μ M) and rapamycin (Rap, 10 nM) were used as positive controls and CuS (0.4 mg/mL) was used as a negative control. iRTX (1 μ M) was used as a TRPV1 antagonist. Laser: 30 cycles. Data are shown as mean \pm S.D. of three independent experiments, and analyzed by Student's *t*-test. * P < 0.05 for CuS-TRPV1 vs. untreated group, # P < 0.05 for Cap vs. untreated group, & P < 0.05 for Rap vs. untreated group.

Comments 1-8:

In Fig. S8, please include CuS and CuS+laser groups to prove the targeting ability of CuS-TRPV1 in lipid reduction.

Response 1-8:

Following the reviewer's suggestion, we investigated the effects of CuS and CuS with laser on lipid accumulation in VSMC. Due to the lack of targeting ability, CuS and CuS with laser failed to impede the oxLDL-induced lipid accumulation and cholesterol increase, as shown in Figure R8 below (Figure S11 in the revised supporting information).

Figure R8. Photothermal activation of TRPV1 by CuS-TRPV1 reduced lipids in VSMCs. VSMCs were cultured in the presence of 80 µg/mL oxLDL for 24 h and treated with capsaicin (Cap, 1 µM) only, CuS-TRPV1 and CuS (0.4 mg/mL) with and without NIR irradiation (980 nm, 5 W/cm², 30 cycles). (a) Representative pictures of Oil red O staining of intracellular lipid droplets. Scale bar = 50 µm. (b) Quantitative analysis of total cholesterol levels. Data are shown as mean ± S.D. of three independent experiments, and analyzed by Student's *t*-test. **P* < 0.05 for CuS-TRPV1 vs. control, #*P* < 0.05 for Cap vs. control.

Comments 1-9:

Please compare Fig. S11 with CuS and quantify their membrane fluorescence intensities.

Response 1-9:

After incubation with non-coated CuS NPs for 24 h, much less fluorescent signal was observed in VSMCs membrane than that with CuS-TRPV1, as shown in Figure R9 below (Figure S14 in the revised supporting information). While it was found that CuS-TRPV1 content does indeed decrease over time (63% that of 2 h incubation) as the cellular internalization, the amount of CuS-TRPV1 remaining on VSMCs membrane was sufficient for generating local heat upon exposure to the laser.

Figure R9. Localization of CuS-TRPV1 and CuS in VSMCs after 24 h incubation. Cells were incubated with fluorescein-conjugated CuS-TRPV1 or CuS (0.4 mg/mL) for 24 h and visualized by IFC. Much less membrane intensity was observed in VSMCs incubated with CuS (blue area: 1,238.72) than that with CuS-TRPV1 (blue area: 17,910.54). CuS-TRPV1 content decreased over time (63% that of 2 h incubation) as the cellular internalization. Data are shown as mean \pm S.D. of three independent experiments, and analyzed by Student's *t*-test. *** $P < 0.001$ for CuS-TRPV1 vs. CuS.

Comments 1-10:

In Fig. 5, why is the CuS-TRPV1 cleared to an initial level at 4 h post-injection rather than remained longer by TRPV1 binding?

Response 1-10:

After intravenous injection of CuS-TRPV1 into mice, these nanoparticles increased cardiovascular-specific PA contrast steadily over time (Fig. 5), achieving a maximum signal-over-noise ratio of 2.94 ± 0.39 at 2 h. Although a large proportion of CuS-TRPV1 accumulated in the cardiovascular, they make direct contact with monocytes and other phagocytic cells in the blood. Once a particle is in contact with these phagocytic cells, receptors on the cell surface either directly recognize the particle, or recognize opsonizing serum proteins attached to the particle. This leads to internalization of the CuS-TRPV1 into the phagocytic cells and rapid clearance from the bloodstream at 4 h post-injection. To further confirm this, *ex vivo* PA imaging was performed as shown in Figure R10 below (Figure S16 in the revised supporting information). After 4 h post-injection, we found increased PA signals in the liver and kidneys due to blood extravasation and filtration by these reticuloendothelial systems.

Figure R10. Biodistribution analysis of CuS-TRPV1 by PA imaging. (a) Representative *ex vivo* PA images of aortic arch and major organs excised at 1, 2, 4 and 6 h post-injection. (b) Quantification of PA signals in aortic arch and each organ. Compared with PA signals of pre-injection, the increased PA signals at indicated time points were calculated. Data are shown as mean \pm S.D. of three independent experiments, and analyzed by Student's *t*-test. * $P < 0.05$ vs. pre in aortic arch, # $P < 0.05$ vs. pre in liver, & $P < 0.05$ vs. pre in kidney.

Comments 1-11:

In Fig. 6, 1) Please add CuS and CuS+laser groups to prove CuS-TRPV1's targeting ability in the therapy; 2) Please provide the western blot analysis and quantification data for pAMPK, LC3I, and LC3II of the aortic arch lesions.

Response 1-11:

Following the reviewer's suggestion, we investigated the therapeutic effects of CuS and CuS plus laser. ApoE^{-/-} mice in these two control groups did not show obvious atherosclerosis suppression as shown in Figure R13 below. (Fig. 6 in the revised manuscript).

Moreover, significant TRPV1 activation was also observed in ApoE^{-/-} mice treated with CuS-TRPV1 + laser and capsaicin, as evidenced by the increased p-AMPK level and LC3II/LC3I ratio in the aortic arch lesions. Only background activation was observed in other groups, which was consistent with the *in vitro* results. The western blots have been quantified and their significance has been confirmed by Student's *t*-test as shown in Figure R11 (Figure S18 in the revised supporting information).

Figure R11. Western blot analysis of p-AMPK and LC3II/ LC3I in the aortic arch lesions from high-fat diet-fed ApoE^{-/-} mice received as indicated treatment. Data are shown as mean ± S.D. of three independent experiments, and analyzed by Student's *t*-test. **P* < 0.05 for CuS-TRPV1 vs. untreated group, #*P* < 0.05 for Cap vs. untreated group.

Minor comments 1-12:

References 10 - 12 are about magnetic nanoparticles. Please clarify terminology.

Response 1-12:

We agree with the reviewer that Ref. 10-12 is only about magnetic nanoparticles. To further support that nanoparticles have emerged as a powerful tool for controlling cell signalling, we have added Ref. 13 and 14 into the revised manuscript. These works are about the application of quantum dots and upconversion nanoparticles for switching of cellular activity and cell signaling.

Minor comments 1-13:

Please include 'synthesis and purification of CuS-TRPV1 and fluorescein conjugated CuS-TRPV1' in the supplementary methods section;

Please include a detailed description of the '*in vivo* photothermal activation study' in the supplementary methods section (eg. number of mice, age, diet detail, therapeutic dose, administration

route, injection frequency, etc.);

Please include a detailed description for the H&E staining of aortic root and the ORO staining in the supplementary methods section.

Response 1-13:

Following the reviewer's suggestion, "the synthesis and purification processes of CuS-TRPV1 and fluorescein conjugated CuS-TRPV1", "*in vivo* photothermal activation study" and "H&E and ORO staining of aortic root" have been discussed in detail in the revised methods and supplementary methods.

Minor comments 1-14:

In Fig. S10, the stability of the CuS-TRPV1 should be evaluated in serum containing media at 37 °C.

Response 1-14:

We are sorry about that the temperature condition was unmentioned. The stability of the CuS-TRPV1 was indeed evaluated in serum containing media at 37 °C.

Response to reviewer 2:

Reviewer 2's comment 2-1:

This work has provided an alternative way to activate TRPV1 in vascular tissue from ApoE knockout mice using NIR-induced heating via CuS nanoparticle-conjugated TRPV1 antibodies. Major finding is that this novel technique could attenuate atherosclerosis. The results sounds interesting and the manuscript is well-written. However, there are still critical issues should be clarified to ensure the safety and reliability of this method.

Response 2-1:

We thank the reviewer for acknowledging the scientific merit of our work. Our responses to the questions about technical issues are listed as following.

Comments 2-2:

VSMCs were incubated with CuS nanoparticle *in vitro* experiments. But *in vivo*, endothelium is the first barrier of arterial wall in response to the stimuli from the blood stream. Why did the authors choose VSMCs as the *in vitro* cellular model rather than endothelial cells? Is there any specificity of CuS nanoparticles binding to VSMCs *in vivo*? The expression of TRPV1 is also abundant in endothelial cells

(Yang DC, *Cell metab*, 2010; Xiong SQ, et al., *Hypertension*, 2016). How did authors exclude the role of endothelial TRPV1 activation in ameliorating atherosclerosis? How did CuS nanoparticle penetrate the plaque containing VSMCs *in vivo*? Authors should provide some experimental evidences for these key points.

Response 2-2:

We are really grateful to the reviewer for inviting us to clarify this important point. Atherosclerosis is a progressive disease that is often considered an inflammatory response to chronic vessel injury, which is mediated in part by perturbed vascular flow and oxidized lipoprotein-mediated vascular inflammation (*Nat Commun.*, 2015, 6: 8995). In the case of vessel injury, endothelial cytokine secretion and leukocyte adhesion molecule upregulation promotes monocyte extravasation from blood vessels into the sub-endothelial space. Subsequently, monocytes differentiate into macrophages, accumulate lipids from the retained lipoproteins, and turn into foam cells. While this is occurring, vascular smooth muscle cells (VSMCs) migrate into and proliferate in the intima. Like macrophages, VSMCs have also been shown to take up lipids and transform into foam cells, contributing to lesion development and progression (*Cardiovasc Res.*, 2012, 95: 165-172; *J. Cell Biol.*, 2015, 209: 13-22). Especially in advanced atherosclerosis lesions, only 30% of foam cells displayed macrophage markers, whereas 45% have a VSMC phenotype. Therefore, VSMCs are an important origin of foam cells besides macrophages, the targeting of which could form an attractive approach to treat atherosclerosis. For these reasons, we choose VSMCs as the *in vitro* cellular model.

TRPV1 channels are also expressed in endothelial cells. Chronic TRPV1 activation by dietary capsaicin increases the activity of protein kinase A (PKA)/ uncoupling protein 2 (UCP2) pathway and endothelial nitric oxide (NO) synthase and thus against ROS-induced endothelial dysfunction and hypertension (*Hypertension*, 201, 67: 451-460; *Cell metab*, 2010, 12: 130-141). No direct evidence implies a role for endothelial TRPV1 in ameliorating atherosclerosis. In contrast, activation of TRPV1 in VSMCs by capsaicin has been shown to significantly reduce vascular lipid accumulation and attenuates atherosclerosis (*Cardiovasc Res.*, 2011, 92: 504-513; *Cell Death Dis.*, 2014, 5: e1182). Therefore, while CuS-TRPV1 likely has actions in multiple cell types *in vivo*, protection from and treatment of atherosclerosis by CuS-TRPV1 mainly contributed to its effects on VSMCs TRPV1.

As we all know, the leakiness in tumor vasculatures leads to penetrate and retention of nanoparticles in the tumor bed which is known as enhanced permeation and retention (EPR) effect (*Nat. Rev. Clin. Oncol.* 2010, 7: 653-664). This EPR effect is also found in atherosclerosis which is associated with the chronic inflammation of arterial blood vessels (*Circulation.* 2004, 110:1330-1336; *Nano Today.* 2014, 9, 223-243). The blood vessels surrounding the atherosclerotic plaques are defective and immature as

that seen in tumor blood vessels. The leaky vasculature allows nanoparticles to pass into the interstitial tissue, while an undeveloped lymphatic drainage system increases accumulation and local concentration of nanoparticles (*Proc. Natl. Acad. Sci. U.S.A.*, 2007, 104: 961-966; *Nano Lett.* 2008, 8: 3715-3723; *Nano Lett.*, 2010, 10: 5131-5138). By virtue of TRPV1-mediated targeting and EPR effect, CuS-TRPV1 could penetrate into the plaques containing VSMCs. To further confirm this, we determine the distribution of CuS-TRPV1 in the aortic arch of mice. Immunofluorescence of aortic arch sections clearly showed that CuS-TRPV1 (shown with green fluorescence of fluorescein) easily penetrated through the arterial wall to the adventitia layer and were trapped by the atherosclerotic plaque containing VSMCs (shown with red fluorescence of α -SMA) at the luminal surface. In contrast, non-coated CuS NPs did not show specific plaque targeting (Figure R12 below, Fig. 5c in the revised manuscript). Thus, CuS-TRPV1 specifically targeted atherosclerotic plaques, concentrating in VSMCs with a desirable consequence.

Figure R12. Representative immunofluorescent micrographs of aortic arch sections demonstrating CuS-TRPV1 within atherosclerotic plaque at 2 h post-injection. From left, fluorescein (green, representing fluorescein-conjugated CuS-TRPRV1 or CuS), α -SMA (red, representing VSMC), DAPI (blue) and merge. The dashed line indicates the plaque border. Scale bar = 100 μ m.

Comments 2-3:

The authors demonstrated that a near-infrared (NIR) light (5 W/cm²) could activate TRPV1 and reduced foam cell formation *in vitro* and attenuated atherosclerosis *in vivo*. It is well-known that temperature greater than 43 °C can be able to activate TRPV1. TRPV1 is also expressed in myocardial,

adipose and endothelial cells. How did the authors determine that the same power of NIR laser (5 W/cm²) could be efficient and accurate to activate the vascular TRPV1 in deep tissues, not involved in TRPV1 in other tissues?

Response 2-3:

As we discussed above (**Response 2-2**), the blood vessels surrounding the atherosclerotic plaque are leaky and exhibit heterogeneous hyperpermeability and defective lymphatic drainage compared to normal tissues. The leaky vasculature increases accumulation and local concentration of CuS-TRPV1 in the plaques. Although TRPV1 also exists in myocardium, adipose and endothelium, penetration of CuS-TRPV1 across the endothelium of the blood vessels into these tissues is hurdle. Because healthy endothelial cells are anchored to a continuous basal membrane and are connected by tight junctions, the continuous endothelium limits the transport of CuS-TRPV1 into these normal tissues. Therefore, NIR laser (5 W/cm²) could be efficient and accurate to activate the vascular TRPV1 in deep tissues without involvement of TRPV1 in other tissues.

To further confirm this, we investigated the NIR activation of TRPV1 signalling in mice. Figure R11 above (Figure S18 in the revised supporting information) showed that the TRPV1 induced-autophagy also existed in the aortic arch lesions. ApoE^{-/-} mice received CuS-TRPV1 treatment with laser irradiation showed significant increased the p-AMPK level and LC3II/LC3I ratio compared with other control groups. The western blots have been quantified and their significance has been confirmed by Student's *t*-test.

Comments 2-4:

In study *in vivo*, mice in the CuS-TRPV1-NIR group was irradiated at the cardiac region with NIR laser (980 nm, 5 W/cm², spot diameter size 0.5 cm). But Figure 6d showed that Enface aortic lesion of those mice was attenuated in the whole aorta, including thoracoabdominal aorta. Obviously, it was beyond the irradiated region. How did authors explain this outcome?

Response 2-4:

Copper (Cu) is an essential trace element in the maintenance of the cardiovascular system. Cu ions are involved in numerous metabolic reactions, forming part of the functional groups of several key enzymes. Among these are enzymes that could be protective against atherosclerosis, including copper-zinc superoxide dismutase (Cu-Zn SOD), and endothelial nitric oxide synthase (eNOS) (*Int J Exp Pathol.*, 2005, 86(4):247-255; *Am J Physiol Endocrinol Metab.*, 2010, 298: E138-E139). Previous reports have shown that dietary copper can reduces the extent of atherosclerosis in the thoracic aorta of cholesterol-fed rabbits (*Atherosclerosis*, 1999, 146: 33-43; *Int. J. Exp. Pathol.*, 2004, 85: 265-275). Since

degradation of inorganic nanoparticles into small molecular complexes is often observed in the physiological environment (*Bioconjugate Chem.* 2015, 26: 511-519; *ACS Nano.*, 2015, 9(7): 6655-6674), we suspected that Cu content within mice might increase after CuS-TRPV1 injection. This would have complementary and synergistic therapeutic effects on atherosclerosis. In the revised manuscript, we carefully analyzed Oil Red O-stained *en face* aortic preparations. Figure. R13 below (Fig. 6 d and f in the revised manuscript) showed that CuS-TRPV1 plus NIR group displayed significant 52.4% reduction in aortic root lesion areas and only 3.3% reduction in thoracoabdominal aortic lesion areas. The obvious reduction mainly located at the cardiac region with NIR irradiation. Although we cannot rule out the possibility that Cu supplements may also ameliorate atherosclerosis, activation of TRPV1 using CuS-TRPV1 should be considered as a key contributor to this therapeutic effect.

Comments 2-5:

The results of Supplementary Fig.13 indicated that administration of capsaicin via intravenous injection caused an obvious weight loss. The authors thought it was the high toxicity of capsaicin. In fact, Zhang et al., reported that the activation of TRPV1 by dietary capsaicin could prevent adipogenesis and obesity safely in mice (Zhang et al., *Cir Res*, 2007). A lot of human study and clinical trials also have shown that spicy food containing capsaicin can prevent obesity. Meanwhile, these differences may be related to the different drug-delivery way. Furthermore, capsaicin is rarely dissolved in water or PBS thus it is usually added in diet to treat animals. I wonder how it was dissolved in PBS and injected intravenously as a positive control in this work.

Response 2-5:

In our study, capsaicin was prepared in absolute ethanol as a 10 mM stock solution and stored at -20 °C. Subsequent dilution was made with PBS to obtain a desired concentration (10 mg/kg) 10 min before intravenously injection. After 12 weeks, an obvious weight decrease (~2.4%) was observed in mice received capsaicin treatment. This outcome might be due, at least partially, to the effect of capsaicin on obesity. However, we should note that two mice in this group died after 70 days. This result might imply that caution should be exercised when long-term topical application of capsaicin. In fact, capsaicin is not generally recognized as safe and effective by the U.S. Food and Drug Administration for fever blister and cold sore treatment, but only considered to be safe and effective as an external analgesic counterirritant. Oral LD50 values as low as 161.2 mg/kg (rats) and 118.8 mg/kg (mice) have been reported for capsaicin in acute oral toxicity studies, with hemorrhage of the gastric fundus observed in some of the animals that died. Intravenous, intraperitoneal, and subcutaneous LD50 values were lower (*J Toxicol Sci.*, 1996, 21(3):195-200). In both oral and intravenous toxicity studies using mice, capsaicin have produced statistically significant differences in the growth rate and

liver/body weight increases (*Int J Toxicol*, 2007, 26(Suppl 1): 3-106). Therefore, although a therapeutic effect of capsaicin on atherosclerosis was obtained in our study, safety concerns should be raised for either capsaicin-containing food supplements or intravenous drugs.

Comments 2-6:

The plaque area in the sections of aortic sinus should be quantified by Oil-red staining. And the significance of the statistical analysis on enface aortic lesions should be labeled.

Response 2-6:

Following the reviewer’s suggestion, atherosclerotic lesions in the aortic sinus and whole aorta were quantified by Oil red O staining and their significance was confirmed by Student’s *t*-test. As shown in Figure. R13 below (Fig. 6 in the revised manuscript), ApoE^{-/-} mice received CuS-TRPV1+NIR treatment displayed a significant 54.1% reduction in aortic root lesion areas ($353 \times 1,000 \mu\text{m}^2$ versus $162 \times 1,000 \mu\text{m}^2$; Fig. 6c, e) and a 55.7% reduction in *en face* prepared aortic lesion areas compared with controls (7.0% versus 3.1%; Fig. 6d, f).

Figure R13. *In vivo* photothermal activation of TRPV1 reduces atherosclerotic lesions in the aorta of ApoE^{-/-} mice on a high-fat diet. (a) NIR laser treatment. (b) In situ aortic arch lesions. Representative images of the plaque at the origins of brachiocephalic artery (BA), left carotid artery (LCA) and left subclavian artery (LSA). (c) Cross-sectional aortic root lesions. Representative images of atherosclerotic lesions in the aortic sinus from consecutive sections of the Oil red O-stained aortic root. Scale bar = 200 μm. (d) *En face* aortic lesions. Representative images of atherosclerotic lesions in Oil Red O-stained aorta. (e) and (f) Quantification of stained area or as a percentage of lesion area. Data are shown as mean ± S.D. of three independent experiments, and analyzed by Student’s *t*-test. **P* < 0.05 for CuS-TRPV1 vs. PBS group, #*P* < 0.05 for Cap vs. PBS group.

Comments 2-7:

Why did the authors choose a frequency twice a week to perform the injection of the nanoparticles and NIR treatment? Could the particles be catalyzed or removed from the circulatory system?

Response 2-7:

To meet the requirements of U.S. Food and Drug Administration (FDA), all injected contrast and therapeutic agents need to be cleared from the body completely in a reasonable time period (*Nat. Biotechnol.* 2007, 25, 1165-1170). The biodistribution and safety of CuS-TRPV1 were further confirmed by *ex vivo* organ inductively coupled plasma mass spectrometry (ICP-AES) for 2 h, 4 h, 24 h and 3 days post-injection as shown in Figure R14 below (Figure S17 in the revised supporting information). By virtue of TRPV1-mediated targeting and EPR effect of atherosclerosis, the accumulation of CuS-TRPV1 in arch reached ~24.6% ID/g at 2 h. On the other hand, low accumulation in the heart, liver, spleen, lung and kidney reflected a relatively low uptake in non-target organs at this time points. Similar to other CuS nanoparticles-based theranostic agents (*ACS Nano.* 2015, 9(7): 6655-6674; *Bioconjugate Chem.* 2015, 26, 511-519), a majority of the intravenously injected CuS-TRPV1 were taken up by the reticuloendothelial system (RES) with liver uptake found to be ~20.4% ID/g at 4 h, and decreased gradually to ~14.2% ID/g at 24 h, suggesting that small CuS nanoparticles can escape macrophage sequestration in RES and might be renally eliminated from the body. The increased uptake in the kidneys at 24 h post-injection (~16.8% ID/g) further confirmed this. After 3 days, the Cu content decreased in most of the tissues and organs. Therefore, to ensure therapeutic efficacy and safety, we choose a frequency twice a week to perform the injection of the nanoparticles and 2 h time point to perform the NIR treatment.

Figure R14. ICP-MS of Cu content at 2 h, 4 h, 24 h and 3 days after intravenous injection with CuS-TRPV1 (0.4 mg/mL). Data are shown as mean \pm S.D. of three independent experiments.

Comments 2-8:

The statistics should be defined in the corresponding figure legends. The diagrams and statistical results should be presented (figure 4b, c). Specify the number of performed experiments in each figure legends. Use symbols (*, #) to show statistical differences in diagrams (figure 6e).

Response 2-8:

Following the reviewer's suggestion, the corresponding quantitative and statistical analysis has been added into the revised manuscript. Each experiment was repeated three times in duplicate if not stated otherwise. Data were presented as mean \pm S.D. Comparisons between groups were analyzed using Student's *t*-test, */#/ & $P < 0.05$ was considered statistically significant.

Response to reviewer 3:

Reviewer 3's comment 3-1:

This paper by Gao et al., describes a nanoparticle approach for treating atherosclerosis using light-activated nanoparticles, to induce a heating effect that activates the TRPV1 channel. The approach

is elegant, and this paper addresses a major unmet therapeutic need. Several issues must be addressed to strengthen the data presented in this manuscript in order to be considered for publication in Nat Comm.

Response 3-1:

We sincerely thank the reviewer for acknowledging the scientific merit of our work. Our responses to the questions about technical issues are listed as following.

Comments 3-2:

First, the *in vivo* data raises some major concerns. Specifically, 1) the authors indicate they performed a single intravenous injection of the targeted nanoparticles. The biodistribution of the particles to the artery must be presented clearly (Figure 5), it is very hard to determine from the images how effective the targeting is. Adding cross-sectional histology of the blood vessel that will indicate where the particles reside, relative the vessel and to the plaque is needed. This will indicate whether the targeted ligand is accessible on the plaque-coated lumen. Furthermore, 2) it seems that one treatment may not be sufficient in order to induce the dramatic therapeutic effect presented 12 weeks post treatment. Some kinetics of the plaque reduction would be reassuring. 3) In addition, one may question the need for a NIR group without nanoparticles to test whether the NIR alone can heat the cells.

Response 3-2:

(1) In the original manuscript, we applied high-frequency ultrasound (US) and photoacoustic (PA) imaging to observe cardiovascular selectivity of CuS-TRPV1 in ApoE^{-/-} mice. US imaging modality provided anatomical features of the aortic arch and its three major branches. PA imaging modality showed that CuS-TRPV1 increased cardiovascular-specific PA contrast steadily over time (Fig. 5a in the revised manuscript), achieving a maximum signal-over-noise ratio of 2.94 ± 0.39 at 2 h. Once a large proportion of CuS-TRPV1 accumulated in the aortic arch, they could penetrate into the plaques containing VSMCs *via* TRPV1-mediated targeting and the EPR effect as discussed above (**Response 2-2**). To further confirm this, we determine the distribution of CuS-TRPV1 in the aortic arch of mice. Immunofluorescence of the aortic arch sections clearly showed that CuS-TRPV1 (shown with green fluorescence of fluorescein) easily penetrated through the arterial wall to the adventitia layer and were trapped by the atherosclerotic plaque containing VSMCs (shown with red fluorescence of α -SMA) at the luminal surface. In contrast, non-coated CuS NPs did not show specific plaque targeting (Figure R12 above, Fig. 5c in the revised manuscript). Thus, CuS-TRPV1 specifically targeted atherosclerotic plaques, concentrating in VSMCs with a desirable consequence.

(2) As we discussed above (**Response 2-7**), a majority of the intravenously injected CuS-TRPV1 were taken up by the reticuloendothelial system (RES) with liver uptake found to be $\sim 20.4\%$ ID/g at 4 h,

and decreased gradually to ~14.2% ID/g at 24 h, suggesting that small CuS nanoparticles can escape macrophage sequestration in RES and might be renally eliminated from the body. The increased uptake in the kidneys at 24 h post-injection (~16.8% ID/g) further confirmed this. After 3 days, the Cu content decreased in most of the tissues and organs. Therefore, to ensure therapeutic efficacy and safety, we choose a frequency twice a week to perform the injection of the nanoparticles and NIR treatment. Moreover, atherosclerosis is a chronic inflammatory disease and its development trends to stabilize in ApoE^{-/-} mice after 12 weeks of high fat-diet (*Arterioscler Thromb Vasc Biol.* 2004, 24:2339-2344). Plaque formation on the aortic sinus and thoracoabdominal aorta can be easily detectable at this time points. To precisely evaluate the therapeutic effect of CuS-TRPV1, lesion size was analyzed histologically after 12 weeks treatment. As shown in Fig. 6 of the revised manuscript, long-term activation of TRPV1 by CuS-TRPV1 plus NIR significantly limited atherosclerotic lesion progression in ApoE^{-/-} mice on a high-fat diet compared to the controls.

(3) Following the reviewer's suggestion, we investigated the therapeutic effect of NIR without nanoparticles. ApoE^{-/-} mice in this control group did not show obvious atherosclerosis suppression as shown in Figure R13 above (Fig. 6 in the revised manuscript).

Comments 3-3:

Regarding the *in vitro* images (Figure 3), confocal z-height images of the nanoparticles on the cells will help better understand if the particles are internalized or on the cell surface.

Response 3-3:

Following the reviewer's suggestion, confocal z-height imaging was further conducted to verify the membrane-targeting ability of CuS-TRPV1. After confirming the cell focusing ranges from the bottom to top, focal plane images were captured at 0.5 μm , 2 μm , 3.5 μm and 5 μm from the bottom. As shown in Figure. R15 (Figure S6 in the revised supporting information), weak fluorescence was found at focal plane near the bottom (0.5 μm from the bottom) as well as focal plane in the middle (2 μm from the bottom), respectively. In contrast, strong green fluorescence signal was observed when we focused at the surface of the cell (5 μm from the bottom). In conjunction with the TEM and IFC images, these results strongly suggest that a high density of CuS-TRPV1 was confined to the plasma membrane after 2 h incubation. Moreover, non-coated CuS nanoparticles did not show such a specific localization on VSMCs membrane, as they were internalized into the interior of cells (strong fluorescence signal at 0.5 μm and 2 μm focal plane).

Figure R15. Representative confocal z-height images of VSMCs incubated with fluorescein conjugated CuS-TRPV1 or CuS (0.4 mg/mL) for 2 h. Scale bar = 5 μm .

Comments 3-4:

Is the activation of the TRPV1 channel alone capable of reversing atherosclerosis or should other pathways be considered as possible contributors to this dramatic therapeutic effect.

Response 3-4:

Signalling pathways that control cholesterol and triglycerides accumulation and foam cells formation are pivotal for atherosclerosis initiation and progression. TRPV1 is a nonselective cation channel, highly permeable to Ca^{2+} , which can be activated by capsaicin and heat. Recently, activation of TRPV1 in VSMCs by capsaicin has been shown to significantly reduce vascular lipid accumulation and attenuates atherosclerosis through autophagy induction (*Cardiovasc Res.*, 2011, 92: 504-513; *Cell Death Dis.*, 2014, 5: e1182). Considering the toxic side effects of capsaicin, here we develop a CuS-TRPV1 switch for photothermal activation of TRPV1 signalling to impede the progression of atherosclerosis. *In vitro* studies showed that activation of TRPV1 by CuS-TRPV1 with NIR irradiation rescued the autophagy impaired by oxLDL via activating Ca^{2+} -AMPK signalling pathway, and ultimately inhibited the VSMC foam cell formation. In the revised manuscript, we further verified the activation of TRPV1-autophagy signaling pathway *in vivo*, seen in the form of increased p-AMPK level and LC3II/LC3I ratio in the aortic arch lesions after CuS-TRPV1 plus NIR treatment. Therefore, the results we present with

CuS-TRPV1 photothermal switch reveals that TRPV1 is a key regulator of vascular lipid accumulation and VSMCs foam cell formation that is required for atherosclerotic plaque development in mice.

Comments 3-5:

Minor remarks:

- 1) Line 182 should read 'chose' (Next we choose high-fat-fed);
- 2) Figure S8b, y-axis should read 'cholesterol'.

Response 3-5:

We are sorry for the typo. In the revised manuscript, we have corrected these errors.

Response to reviewer 4:

Reviewer 4's comment 4-1:

In this study entitled "CuS Nanoparticle as a photothermal switch for the control of TRPV1 signalling to attenuate atherosclerosis" Gao and colleagues develop a method that allow them to activate TRPV1 channels in the artery wall by using a near-infrared (NIR) red to activate TRPV1 channels and attenuate atherosclerosis in mice. Overall, the concept is of interest. However, the results do not support author's conclusions and the characterization of plaque lesions is poorly done. It is recommended that an expert in atherosclerotic studies supervise this study.

Response 4-1:

We have conducted supplementary experiments to further support our conclusions, and revised the manuscript substantially according to the comments from all referees. We hope the revised paper could satisfy the reviewer.

Comments 4-2:

- (1) There is not quantification of plaque size in the aortic root (Fig. 6c). Why the authors overlook highlighting the biggest plaque in the CuS-TRPV1+laser group???
- (2) If activation of laser reduces lipid deposition, the authors should assess neutral lipid deposition (Oil Red O staining) in the cross-sectional analysis (aortic root).

Response 4-2:

Following the reviewer's suggestion, atherosclerotic lesions in the aortic root and whole aorta were quantified by Oil red O staining and their significance was confirmed by Student's *t*-test. As shown in

Figure. R13 above (Fig. 6 in the revised manuscript), ApoE^{-/-} mice received CuS-TRPV1+NIR treatment displayed a significant 54.1% reduction in aortic root lesion areas ($353 \times 1,000 \mu\text{m}^2$ versus $162 \times 1,000 \mu\text{m}^2$; Fig. 6c, e) and a 55.7% reduction in *en face* prepared aortic lesion areas compared with controls (7.0% versus 3.1%; Fig. 6d, f).

Comments 4-3:

How the laser is applied, it looks that influence lipid deposition in the whole aorta (enface plaque analysis).

Response 4-3:

We are really grateful to the reviewer for inviting us to clarify this point. ApoE^{-/-} mice in the CuS-TRPV1+NIR group were irradiated using NIR laser light (980 nm, 5 W/cm², spot diameter size 0.5 cm, Fig. 6a) at the cardiac region for 30 cycles. But Fig. 6d in the original manuscript showed that *en face* aortic lesion of those mice was attenuated in the whole aorta, also including abdominal aorta. We speculate that this may be due to the synergistic suppression effect of copper (Cu) ions on atherosclerosis. As we discussed in **Response 2-4**, copper (Cu) ions are involved in numerous metabolic reactions, forming part of the functional groups of several key enzymes. Among these are enzymes that could be protective against atherosclerosis, including copper-zinc superoxide dismutase (Cu-Zn SOD), and endothelial nitric oxide synthase (eNOS) (*Int J Exp Pathol.*, 2005, 86(4):247-255; *Am J Physiol Endocrinol Metab.*, 2010, 298: E138-E139). Previous reports have shown that dietary copper can reduce the extent of atherosclerosis in the thoracic aorta of cholesterol-fed rabbits (*Atherosclerosis*, 1999, 146: 33-43; *Int. J. Exp. Pathol.*, 2004, 85: 265-275). Since degradation of inorganic nanoparticles into small molecular complexes is often observed in the physiological environment (*Bioconjugate Chem.* 2015, 26: 511-519; *ACS Nano.*, 2015, 9(7): 6655-6674), we suspected that Cu content within mice might increase after CuS-TRPV1 injection. This would have complementary and synergistic therapeutic effects on atherosclerosis. In the revised manuscript, we further analyzed Oil Red O-stained *en face* aortic preparations. Fig. 6e and Fig. 6f (in the revised manuscript) showed that CuS-TRPV1 plus NIR group displayed a significant 52.4 % reduction in aortic root lesion areas and a 3.3% reduction in thoracoabdominal aortic lesion areas. The obvious reduction mainly located at the cardiac region with NIR irradiation. Although we cannot rule out the possibility that Cu supplements may also ameliorate atherosclerosis, activation of TRPV1 using CuS-TRPV1 should be considered as a key contributor to this therapeutic effect.

Comments 4-4:

The authors should also demonstrate that transient activation of TRPV1 influences p-AMPK and autophagy *in vivo*.

Response 4-4:

Following the reviewer's suggestion, we investigate whether activation of TRPV1 by CuS-TRPV1+NIR could induce AMPK phosphorylation and autophagy *in vivo*. Figure R11 (Figure S18 in the revised supporting information) showed that this induction existed in the aortic arch lesions. ApoE^{-/-} mice received CuS-TRPV1 treatment with laser irradiation showed significant increased the p-AMPK level and LC3II/LC3I ratio compared with other control groups. The western blots have been quantified and their significance has been confirmed by Student's *t*-test.

Reviewers' comments:

Reviewer #1 (Remarks to the Author):

The manuscript by Wen Gao et al. entitled, "CuS nanoparticle as a photothermal switch for the control of TRPV1 signaling to attenuate atherosclerosis", describes remote controlled nanotherapeutics capable of selective stimulation of transient receptor potential vanilloid subfamily 1 (TRPV1) located on the plasma membrane of vascular smooth muscle cell (VSMC) in atherosclerosis. Near infrared (NIR) absorbing CuS-TRPV1 was designed to provide photoacoustic (PA) image-guided therapy as well as to stimulate thermosensitive TRPV1 in a controlled manner around the atherosclerotic plaques for autophagy activation and subsequent reduction in lipid accumulation. Remote regulation of cellular activity using inorganic nanoparticles is a potentially important technique for elucidating underlying physiological processes with promising therapeutic potential (Nature medicine 21; 92-98, 2015). A recent report on wireless magnetothermal deep brain stimulation shows remote neural excitation through magnetic nanoparticle-mediated TRPV1 activation (Science 347; 1477-1480, 2015). The authors have provided additional data and answered to the reviewer's comments. The author's point-by-point response and the revised manuscript have addressed major issues original reviewers raised. The following are additional minor comments, which should be addressed prior to the acceptance of the manuscript in Nature Communication:

1. In Figure S16, please include each organ's name in the figure.
2. Figure S18a is a critical data which demonstrates CuS-TRPV1 treatment with laser irradiation can increase p-AMPK level, subsequently upregulate LC3II/LC3I ratio. However, western images of the figure look different when compared with other western results in the manuscript. Why do the western images of p-AMPK (laser- , CuS + laser+, CuS-TRPV1 + laser-) show blur white lines? Please add description to clarify the data.
3. Please include quantification method of the western blot results in the method section.

Reviewer #2 (Remarks to the Author):

In the current version of manuscript, the authors have performed additional experiments to support the conclusion. However, as the quality of some results are not high enough to be shown. Thus, some experiments should be repeated and more solid data need to provide.

1. In the newly added Figure 5C (or Figure R12), the fluorescent staining results for fluoresce and α -SMA are not specific. A negative control IgG should be added to ensure the staining was not background signal.
2. In the figure R5, iRTX was not labeled in the figure, and the immunoblot staining of actin in samples "laser" and "laser+CuS-TRPV1" are significantly lower than other groups. And p-AMPK should be normalized to total AMPK rather than actin.
3. The quality of all western blots is poor and need to be greatly improved for statistical analysis.
4. The comparison of the AS reduction rate between aortic root region and thoracic-abdominal aortic areas could not be found in Figures 6d and 6f.
5. Although authors explain the toxicity of capsaicin based on literature, lack of convinced experimental data supports the two mice died of capsaicin toxicity in this study. How do you exclude the other reasons? Do you have any pathological evidences to support your results?
6. A lot of experimental and human studies have shown the beneficial effects of dietary capsaicin and spicy food on cardiometabolic diseases (Please see Sun F, et al. Nutrients. 2016 Apr 25;8(5). pii: E174. doi: 10.3390/nu8050174). What is advantage of your method compared with dietary capsaicin intervention? Authors should explain this point in the discussion

Reviewer #3 (Remarks to the Author):

The authors have addressed the issues I was concerned of.

Reviewer #4 (Remarks to the Author):

While the authors have addressed some of my concerns. I still have some important issues about this study.

- 1- The atherosclerosis analysis is performed using a limited number of mice (n=3-4). This is unacceptable in the field. The authors should include at least 10 mice per group for individual experiments.
- 2- The representative aortas showed in the en face analysis (Fig 6D, aortas 3-7) do not correlate with the ORO qualification in right panel.
- 3- I am still have some concerns with the effect of Cu. The authors should perform additional quantitative analysis to determine whether the photothermal activation in the aortic arch attenuates atherosclerosis. If the photo thermal activation influences atherogenesis in mice injected with CuS-TRPV1, the magnitude of atheroprotection should be greater in the aortic arch compared to thoracic aorta.
- 4- The authors claim that the inhibition of foam cell formation is due to enhance autophagy. However, the authors should test this mechanistically by assessing cellular cholesterol efflux, which is affected by autophagy. This part is very descriptive.
5. The staining of VSMC and CuS-TRVP1 is not convincing. VSMC are usually located in the media or in the fibrous cap of the atherosclerotic lesions. Additional analysis and quantification is essential to demonstrate the specificity of CuS-TRVP1 I for VSMC.

Point-by-point response to the reviewer's comments

Response to reviewer 1:

Overall Comments: The manuscript by Wen Gao et al. entitled, “CuS nanoparticle as a photothermal switch for the control of TRPV1 signaling to attenuate atherosclerosis”, describes remote controlled nanotherapeutics capable of selective stimulation of transient receptor potential vanilloid subfamily 1 (TRPV1) located on the plasma membrane of vascular smooth muscle cell (VSMC) in atherosclerosis. Near infrared (NIR) absorbing CuS-TRPV1 was designed to provide photoacoustic (PA) image-guided therapy as well as to stimulate thermosensitive TRPV1 in a controlled manner around the atherosclerotic plaques for autophagy activation and subsequent reduction in lipid accumulation. Remote regulation of cellular activity using inorganic nanoparticles is a potentially important technique for elucidating underlying physiological processes with promising therapeutic potential (Nature medicine 21; 92-98, 2015). A recent report on wireless magnetothermal deep brain stimulation shows remote neural excitation through magnetic nanoparticle-mediated TRPV1 activation (Science 347; 1477-1480, 2015). The authors have provided additional data and answered to the reviewer's comments. The author's point-by-point response and the revised manuscript have addressed major issues original reviewers raised. The following are additional minor comments, which should be addressed prior to the acceptance of the manuscript in Nature Communication.

Comment 1-1: In Figure S16, please include each organ's name in the figure.

Response 1-1: We much appreciate the reviewer's encouraging and constructive comments. Following the reviewer's suggestion, we have added each organ's name in Figure R1 below and Figure S17 in the revised supporting information.

Figure R1. Biodistribution analysis of CuS-TRPV1 by PA imaging. (a) Representative *ex vivo* PA images of aortic arch and major organs excised at 1, 2, 4 and 6 h post-injection. (b) Quantification of PA signals in aortic arch and each organ. Compared with PA signals of pre-injection, the increased PA signals at indicated time points were calculated. Data are shown as mean \pm S.D. (n = 3), and analyzed by Student's *t*-test. **P* < 0.05 vs. pre in aortic arch, #*P* < 0.05 vs. pre in liver, &*P* < 0.05 vs. pre in kidney.

Comment 1-2: Figure S18a is a critical data which demonstrates CuS-TRPV1 treatment with laser irradiation can increase p-AMPK level, subsequently upregulate LC3II/LC3I ratio. However, western images of the figure look different when compared with other western results in the manuscript. Why do the western images of p-AMPK (laser-, CuS + laser+, CuS-TRPV1 + laser-) show blur white lines? Please add description to clarify the data.

Response 1-2: We greatly appreciate the reviewer's kind remarks. The main reason that western images of p-AMPK (laser-, CuS + laser+, CuS-TRPV1 + laser-) showed blur white lines may be due to the higher concentration of primary or secondary antibody used in the blotting experiment. To correct for this, the experiment was done using a lower concentration of antibodies (1:1200) in the revised manuscript. Representative western images were listed in Figure R2 below (Figure S20 in the revised supporting information). The p-AMPK level and LC3II/LC3I ratio were quantified by the western blots.

Figure R2. Western blot analysis of p-AMPK/AMPK and LC3II/ LC3I in the aortic arch lesions from high-fat diet-fed ApoE^{-/-} mice received as indicated treatment. Data are shown as mean \pm S.D. (n = 6) and analyzed by Student's *t*-test. **P* < 0.05 for CuS-TRPV1 vs. untreated group, #*P* < 0.05 for Cap vs.

untreated group.

Comment 1-3: Please include quantification method of the western blot results in the method section.

Response 1-3: Following the reviewer's suggestion, "quantification method of the western blot results" has been added to the method section of revised manuscript.

Response to reviewer 2:

Overall Comments: In the current version of manuscript, the authors have performed additional experiments to support the conclusion. However, as the quality of some results are not high enough to be shown. Thus, some experiments should be repeated and more solid data need to provide.

Comment 2-1: In the newly added Figure 5C (or Figure R12), the fluorescent staining results for fluoresce and α -SMA are not specific. A negative control IgG should be added to ensure the staining was not background signal.

Response 2-1: We much appreciate the reviewer's helpful comments. Following the reviewer's advice, we have repeated some experiments, as well as performed supplementary experiments to strengthen our data. To better evaluate the specificity of α -SMA, sequential sections of aortic arch, including both atherosclerotic and healthy aortas, were stained for α -SMA and IgG, respectively. Immunofluorescence micrographs of atherosclerotic aortas (Figure R3 below, Figure 5c-d in the revised manuscript, Figure S18 in the revised supporting information) revealed significant levels of α -SMA protein (red fluorescence) that colocalized with CuS-TRPV1 (green fluorescence), whereas there was no IgG staining that colocalized with CuS-TRPV1. These results demonstrated that fluorescence and α -SMA are specific and staining was not background signal. Little green fluorescence observed in healthy aortas further confirmed the plaque targeting of CuS-TRPV1.

Figure R3. (a) Representative immunofluorescent micrographs of aortic root sections showing CuS-TRPV1 accumulation in atherosclerotic (upper panel) and healthy (lower panel) aortas at 2 h post-injection. From right, blue (DAPI), red (VSMC maker α -SMA or background control IgG), green (fluorescein-conjugated CuS-TRPV1) and merge. The white dashed square gave a view of the magnification on the right (Scale bar = 50 μ m). The red line indicated the lesion border. (b) and (c) Fluorescence intensity profiles of the regions enclosed by the white dotted lines.

Comments 2-2: In the figure R5, iRTX was not labeled in the figure, and the immunoblot staining of actin in samples “laser” and “laser+CuS-TRPV1” are significantly lower than other groups. And p-AMPK should be normalized to total AMPK rather than actin.

Response 2-2: We much appreciate the reviewer’s helpful comments. Following the reviewer’s advice, iRTX has been labeled. Western blots for total AMPK and p-AMPK levels have been reperformed *in vitro* and *in vivo*. The relative levels of AMPK phosphorylation were normalized to total AMPK, with β -actin serving as a loading control. The corresponding western images were listed in Figure R2 above, Figure R4 and R5 below (Figure S20, S9 and 3d in the revised supporting information and manuscript).

Figure R4. Western blot analysis of the involvement of TRPV1 in AMPK phosphorylation. VSMCs were incubated with CuS-TRPV1 (0.4 mg/mL) combined with iRTX (1 μ M) and irradiated by the 980 nm laser (5 W/cm²) for 10 cycles. Capsaicin (Cap, 1 μ M) was used as a positive control and CuS (0.4 mg/mL) was used as a negative control. The relative levels of AMPK phosphorylation were normalized to total AMPK, with β -actin serving as a loading control. Data are shown as mean \pm S.D. of three independent experiments, and analyzed by Student's *t*-test. No statistical significance was detected.

Figure R5. Western blot analysis of AMPK phosphorylation in VSMCs induced by CuS-TRPV1 heating-evoked Ca^{2+} influx. VSMCs were incubated with 0.4 mg/mL CuS-TRPV1 and irradiated by the 980 nm laser (5 W/cm^2) for 10 cycles. Capsaicin (Cap, 1 μM) was used as a positive control and CuS (0.4 mg/mL) was used as a negative control. The relative levels of AMPK phosphorylation were normalized to total AMPK, with β -actin serving as a loading control. Data are shown as mean \pm S.D. of three independent experiments, and analyzed by Student's *t*-test. ** $P < 0.01$ for CuS-TRPV1 + laser vs. untreated group. # $P < 0.05$ for Cap vs. untreated group.

Comments 2-3: The quality of all western blots is poor and need to be greatly improved for statistical analysis.

Response 2-3: Following the reviewer's suggestion, we repeated all western blots and improved the corresponding statistical analysis. The western blot results with its quantification method have been discussed in detail in the revised manuscript and supporting information.

Comments 2-4: The comparison of the AS reduction rate between aortic root region and thoracic-abdominal aortic areas could not be found in Figures 6d and 6f.

Response 2-4: To confirm that atherosclerotic reduction was mainly located at the cardiac region with CuS-TRPV1 plus NIR irradiation, we carefully analyzed Oil Red O-stained *en face* aortic preparations of 70 ApoE^{-/-} mice (n = 10/group) using NIS-Elements imaging software (Nikon, Japan). The whole

aorta was divided into two subcomponents, aortic arch and thoracic-abdominal aorta. The corresponding percentage of lesion area was calculated as Oil Red O-stained area divided by total aorta surface area and the results were listed in Figure R6 below (Figure 6c, e and f in the revised manuscript). At the aortic arch region with NIR irradiation (within the dashed box above), photothermal activation by CuS-TRPV1 resulted in a significant 72.1% reduction (3.1% vs. 11.1%) in aortic arch lesion areas compared with PBS group. At the thoracic-abdominal aorta region outside NIR irradiation (within the dashed box below), only 7.2% reduction (0.69% vs. 0.64%) was obtained. The drastic difference in lesion reduction between the aortic arch and thoracic-abdominal aorta indicates that CuS-TRPV1 plus NIR irradiation had atheroprotective abilities specifically targeting cardiac region.

Figure R6. (a) Representative photographs of *en face* preparations of aortas stained with Oil Red O. (b) and (c) Quantification of stained area as a percentage of whole aorta. Data are shown as mean \pm S.D. (n = 10), and analyzed by Student's *t*-test. ** $P < 0.01$ for CuS-TRPV1 + laser vs. PBS, # $P < 0.05$ for Cap vs. PBS, & $P < 0.05$ for CuS-TRPV1 + laser vs. Cap.

Comments 2-5: Although authors explain the toxicity of capsaicin based on literature, lack of convinced experimental data supports the two mice died of capsaicin toxicity in this study. How do you exclude the other reasons? Do you have any pathological evidences to support your results?

Response 2-5: To exclude the potential cause of death being intravenous toxicity as suggested by the reviewer, capsaicin was administered intragastrically at a concentration of 10 mg/kg twice a week in the revised manuscript. One minor observation made after the administration process was that the mice had slower recoveries to normal daily function and a decrease in food intake. After 12 weeks of treatment, a toxicity study of capsaicin was performed by haematoxylin and eosin (H&E) staining of major organs and monitoring changes in body weight. Figure R7 below (Figure S25 in the revised supporting information) presented obvious liver damage in capsaicin group, as evidenced by cellular shrinkage and steatosis in liver cells. In addition, as shown in Figure R8 below (Figure S24 in the revised supporting information), a significant weight difference between the capsaicin group (25.8 g) and other groups (33.0~33.8 g) was observed. This may be due to the noticeable decrease in food intake, indicating the possibility of irritation to the gastrointestinal tract. These results provided strong pathological evidences of the toxic potential of capsaicin.

Figure R7. Representative histology (H&E) images of major organs collected from treatment (laser; CuS; CuS+laser; CuS-TRPV1; CuS-TRPV1 + laser; Capsaicin) and control (PBS only) groups of mice after 12 weeks (Scale bar = 100 μ m). Inset: magnification of hepatocellular steatosis and shrinkage (Scale bar = 10 μ m).

Figure R8. 12-week growth chart of ApoE^{-/-} mice on a high-fat diet from treatment (laser; CuS; CuS + laser; CuS-TRPV1; CuS-TRPV1 + laser; capsaicin) and control (PBS only) groups (n = 10).

Comments 2-6: A lot of experimental and human studies have shown the beneficial effects of dietary capsaicin and spicy food on cardiometabolic diseases (Please see Sun F, et al. *Nutrients*. 2016 Apr 25;8(5). pii: E174. doi: 10.3390/nu8050174). What is advantage of your method compared with dietary capsaicin intervention? Authors should explain this point in the discussion

Response 2-6: Thanks for the reviewer's comments. In the revised manuscript, atherosclerotic lesion analysis was reperformed using total of 70 ApoE^{-/-} mice (n = 10/group) and the results were listed in Figure R6 above (Figure 6, S22 and S23 in the revised manuscript and supporting information). Because CuS-TRPV1 switch can confine its therapeutic effects to the cardiac region, a more effective therapeutic outcome was obtained as assessed by Oil red O staining compared to TRPV1 agonists capsaicin. This serves an important purpose because it can target treatment of areas prone to atherosclerotic lesions, such as aortic arch, carotid artery and femoral artery, which cannot be achieved by capsaicin alone. More importantly, long-term activation of TRPV1 by our switch showed no obvious *in vivo* toxicity. In virtue of these advantages, the developed CuS-TRPV1 switch has the potential to be a powerful tool for accurate image-guided therapy of atherosclerosis. Detailed discussion has been added to the final paragraph in the results, as well as the discussion.

Response to reviewer 3:

Reviewer 3's comment 3-1: The authors have addressed the issues I was concerned of.

Response 3-1: We are happy to hear that the reviewer was satisfied with the revisions that have been done. Thanks again for reviewing the manuscript.

Response to reviewer 4:

Overall Comments: While the authors have addressed some of my concerns. I still have some important issues about this study.

Response: We have conducted supplementary experiments to further support our conclusions, and our responses to the questions about technical issues are listed as follows.

Comments 4-1: The atherosclerosis analysis is performed using a limited number of mice (n = 3-4). This is unacceptable in the field. The authors should include at least 10 mice per group for individual experiments.

Response 4-1: Thanks for the reviewer's helpful comments. In the revised manuscript, a total of 121 ApoE^{-/-} mice (male, 6-8 weeks old) were fed with normal diet (ND, n = 3), or high-fat diet (HFD, 20% fat, 20% sugar, and 1.25% cholesterol, n = 118) for 12 weeks. HFD-fed mice were randomly divided into seven groups: (a) control group (n = 22); (b) laser group (n = 16); (c) CuS group (n = 16) (d) CuS + laser group (n = 16); (e) CuS-TRPV1 group (n = 16); (f) CuS-TRPV1 + laser group (n = 16); (g) capsaicin group (n = 16). Three ND-fed mice and six control group mice were used for confirming *in vivo* specificity of CuS-TRPV1 for plaque VSMCs. Six mice from each HFD group were used for western blot analysis of p-AMPK/AMPK, LC3II/LC3I and ABCA1 expression and the total cholesterol level in aortic arch. Ten mice from each HFD group were used for evaluation of atherosclerotic lesion. Same results were collected as in the previous experiments using three to four mice. The increased number of the mice per experimental group further consolidates previous conclusion and ensures reliability of the data.

Comments 4-2: The representative aortas showed in the *en face* analysis (Fig 6D, aortas 3-7) do not correlate with the ORO qualification in right panel.

Response 4-2: As we mentioned above, atherosclerotic lesion analysis was reperformed using ten mice from each HFD group. Oil Red O-stained *en face* aortic preparations (Figure S23 in the revised supporting information) were carefully analyzed using NIS-Elements imaging software (Nikon, Japan). The whole aorta was divided into two subcomponents, aortic arch and thoracic-abdominal aorta. The

corresponding percentage of lesion area was calculated as Oil Red O-stained area divided by total aorta surface area. Data in Figure R6b and c (Figure 6e and f in the revised manuscript) are shown as mean \pm S.D. (n = 10), and analyzed by Student's *t*-test, which directly correlate with the representative aortas shown in Figure R6a (Figure 6c in the revised manuscript).

Comments 4-3: I still have some concerns with the effect of Cu. The authors should perform additional quantitative analysis to determine whether the photothermal activation in the aortic arch attenuates atherosclerosis. If the photothermal activation influences atherogenesis in mice injected with CuS-TRPV1, the magnitude of atheroprotection should be greater in the aortic arch compared to thoracic aorta.

Response 4-3: Thanks for the reviewer's helpful comments. To determine whether the photothermal activation in the aortic arch attenuates atherosclerosis, we carefully analyzed Oil Red O-stained *en face* aortic preparations using NIS-Elements imaging software (Nikon, Japan) as we showed in Figure R6 above (Figure 6c, e, f in the revised manuscript). At the aortic arch region with NIR irradiation (within the dashed box above), photothermal activation by CuS-TRPV1 resulted in a significant 72.1% reduction (3.1% vs. 11.2%) in aortic arch lesion areas compared with PBS group. At the thoracic-abdominal aorta region outside NIR irradiation (within the dashed box below), only 7.2% reduction (0.64% vs. 0.69%) was obtained. The drastic difference in lesion reduction between aortic arch and thoracic-abdominal aorta indicates that CuS-TRPV1 plus NIR irradiation had atheroprotective abilities specifically targeting the cardiac region. In addition, to rule out the possible therapeutic effect of Cu, comparative analysis was done between CuS-TRPV1 without NIR irradiation (aortic arch: 11.1%; thoracic-abdominal aorta: 0.69%) and PBS group (aortic arch: 11.2%; thoracic-abdominal aorta: 0.69%) with no significant difference. These results showcase the measurable value of photothermal activation in the treatment of atherosclerosis.

Comments 4-4: The authors claim that the inhibition of foam cell formation is due to enhance autophagy. However, the authors should test this mechanistically by assessing cellular cholesterol efflux, which is affected by autophagy. This part is very descriptive.

Response 4-4: We greatly appreciate the reviewer's helpful comments. It has been demonstrated that the delivery of lipid droplets to lysosomes occurs via autophagy. Upon delivery, lysosomal acid lipase acts to hydrolyze lipid droplet cholesterol esters to generate free cholesterol mainly for ATP-binding cassette transporter A1 (ABCA1)-dependent efflux (*Cell Metab* 2011; 13: 655-667). Because of this, activation of TRPV1 induced autophagy has a potential role in promoting cholesterol efflux and regulating foam cell formation in atherosclerosis. In the original manuscript, the impaired autophagy

was markedly rescued by CuS-TRPV1+laser through formation of numerous autophagosomes and increase of LC3-II/LC3-I ratio, which was revealed by TEM and WB *in vitro* and *in vivo*, respectively. In the revised manuscript, we further verified the connection between autophagy and atherogenesis by assessing cellular cholesterol efflux. As shown in Figure R9 below (Figure S12 in the revised supporting information), autophagy enhancement significantly upregulated ABCA1 expression and increased cholesterol efflux, leading to the inhibition of VSMC foam cell formation. *In vivo* experiments also showed significantly higher expression of ABCA1 in aortic arch of ApoE^{-/-} mice in CuS-TRPV1+laser group compared to PBS group. As a result, a lower total cholesterol level was observed (Figure R10 below, Figure S21 in the revised supporting information). Collectively, these experimental results demonstrated that attenuation of atherogenesis by means of CuS-TRPV1 heating is due to autophagy-promoted ABCA1-mediated cholesterol efflux.

Figure R9. Photothermal activation of TRPV1 by CuS-TRPV1 upregulated ABCA1 expression and increased cholesterol efflux in VSMCs. OxLDL (80 $\mu\text{g}/\text{mL}$) or fluorescent-labeled cholesterol (5 μM)-pretreated VSMCs were incubated with CuS-TRPV1 (0.4 mg/mL) and irradiated by the 980 nm laser (5 W/cm^2) for 30 cycles. Capsaicin (Cap, 1 μM) was used as a positive control and CuS (0.4 mg/mL) was used as a negative control. (a) Western blot analysis of ABCA1 expression after indicated treatments. β -actin was used as a loading control. (b) Quantitative analysis of cholesterol efflux after indicated treatments. Data are shown as mean \pm S.D. of three independent experiments, and analyzed by Student's *t*-test. * $P < 0.05$ for CuS-TRPV1 + laser vs. untreated group, # $P < 0.05$ for Cap vs. untreated group.

Figure R10. (a) Western blot analysis of ABCA1 expression and (b) Quantitative analysis of total cholesterol levels in the aortic arch lesions from high-fat diet-fed ApoE^{-/-} mice received as indicated treatment. Data are shown as mean \pm S.D. (n = 6), and analyzed by Student's *t*-test. **P* < 0.05 for CuS-TRPV1 vs. untreated group, #*P* < 0.05 for Cap vs. untreated group.

Comments 4-5: The staining of VSMC and CuS-TRPV1 is not convincing. VSMC are usually located in the media or in the fibrous cap of the atherosclerotic lesions. Additional analysis and quantification is essential to demonstrate the specificity of CuS-TRPV1 for VSMC.

Response 4-5: We much appreciate the reviewer's helpful comments. Following the reviewer's advice, we have reperformed immunofluorescent staining, as well as done additional analysis and quantification to strengthen our data. To better evaluate the specificity of CuS-TRPV1 for VSMC, sequential sections of aortic arch, including both atherosclerotic and healthy aortas, were stained for α -SMA and IgG, respectively (Figure R11 below, Figure 5c-e in the revised manuscript, Figure S18 in the revised supporting information). Compared to IgG background control, immunofluorescence micrographs of atherosclerotic aortas revealed significant levels of CuS-TRPV1 (green fluorescence) that overlaid with the VSMC marker α -SMA (red fluorescence). In healthy aortas, there was little CuS-TRPV1 that colocalized with α -SMA. This is mainly because the blood vessels surrounding the atherosclerotic plaques are defective and immature (*Circulation*. 2004, 110:1330-1336; *Nano Today*. 2014, 9, 223-243). The leaky vasculature surrounding the atherosclerotic plaques allows CuS-TRPV1 to easily penetrate the arterial wall and be trapped by the atherosclerotic plaque containing VSMCs.

Because early plaque progression involves migration of VSMCs from the media to the intima, proliferation of resident intimal and media-derived VSMCs and formation of VSMC foam cells (Figure 1 in *Nature*. 2011, 473: 317-323), the colocalization of red fluorescence of α -SMA and green fluorescence of CuS-TRPV1 was mainly found at intimal and media layers. Without TRPV1-mediated targeting, non-coated CuS NPs did not show such specific localization in plaque VSMCs. These results together demonstrated the plaque-targeting and penetrating ability of CuS-TRPV1.

Figure R11. (a) Representative immunofluorescent micrographs of aortic root sections demonstrating CuS-TRPV1 within atherosclerotic plaque at 2 h post-injection. From right, blue (DAPI), red (VSMC maker α -SMA or background control IgG), green (fluorescein-conjugated CuS-TRPV1 or CuS) and merge. The white dashed square gave a view of the magnification on the right (Scale bar = 50 μ m). The red line indicated the lesion border. (b-d) Fluorescence intensity profiles of the regions enclosed by the white dotted lines.

REVIEWERS' COMMENTS:

Reviewer #1 (Remarks to the Author):

The authors have addressed my critiques in their revised manuscript, and I support the acceptance of this manuscript in its current form.

Reviewer #2 (Remarks to the Author):

1. The capsaicin was administered intragastrically (group g) at a concentration of 10 mg/kg. This would result in an intense stimulation of gastrointestinal tubes. In addition, the authors did not mention how capsaicin was dissolved. If it was dissolved in ethanol, the liver damage could be possibly a result of excessive use of ethanol but not capsaicin itself. The authors should clarify this issue.

Reviewer #3 (Remarks to the Author):

In my opinion the authors respond in an adequate manner to the reviewer's concerns. In light of the above I support publishing this paper in its current form.

Reviewer #4 (Remarks to the Author):

The authors have addressed my previous concerns.

Response to reviewer 2:

Comment: The capsaicin was administered intragastrically (group g) at a concentration of 10 mg/kg. This would result in an intense stimulation of gastrointestinal tubes. In addition, the authors did not mention how capsaicin was dissolved. If it was dissolved in ethanol, the liver damage could be possibly a result of excessive use of ethanol but not capsaicin itself. The authors should clarify this issue.

Response: Thanks for the reviewer's comments. In our study, capsaicin was dissolved in absolute ethanol at a concentration of 40 mg mL⁻¹ in stock solution and stored at -20 °C. The capsaicin was administered intragastrically (group g) at a concentration of 10 mg kg⁻¹. Based on the body weight of mice (~20-26 g), 5-6.5 µL of capsaicin stock solution was diluted to 150 µL with PBS. Because no more than 6.5 µL of ethanol was administered intragastrically to each mouse, we deduced that the liver damage could not have been due to excess ethanol, but rather capsaicin itself. To further confirm this, we use the same concentration of absolute ethanol/PBS solution without capsaicin as a control. The mice treated with the control showed normal daily function with no impact on food intake after administration.

To address the reviewer's concern with the administration method, our choice of intragastric drug administration and 10 mg kg⁻¹ capsaicin concentration was referenced from previous studies (*Biosci Biotechnol Biochem.* 2009, **73**, 1021-1027; *Cardiovasc Res.* 2011, **92**, 504-513), which utilized similar methods and concentrations. A concentration of 10 mg kg⁻¹ of capsaicin has been shown in previous studies using mouse models to be the minimum effective dosage in the activation of TRPV1 and treatment of atherosclerosis. In our study, this concentration was administered in two separate doses each week, further reducing the potential of irritation to the gastrointestinal tubes of the mice. As well, passive reflex, a common concern with this form of administration was not observed during the oral gavage procedure. Oral gavage was conducted by a skilled technician to ensure proper mice handling and to avoid complications. Furthermore, intragastric drug administration is a frequently used method in animal experiments that is deemed safe and non-invasive. Based on these indications, the method used in our study should not result in intense stimulation of the gastrointestinal tubes, and thus should not affect the accuracy and validity of the results.